# Variance estimation in compound decision theory under boundedness

**Subhodh Kotekal**
Department of Statistics
University of Chicago
Chicago, IL 60637
skotekal@uchicago.edu

## Abstract

The normal means model is often studied under the assumption of a known variance. However, ignorance of the variance is a frequent issue in applications and basic theoretical questions still remain open in this setting. This article establishes that the sharp minimax rate of variance estimation in square error is $(\log \log n / \log n)^2$ under arguably the most mild assumption imposed for identifiability: bounded means. The rate-optimal estimator proposed in this article achieves the optimal rate by estimating $O(\log n / \log \log n)$ cumulants and leveraging a variational representation of the noise variance in terms of the cumulants of the data distribution. The minimax lower bound involves a moment matching construction.

## 1   Introduction

Consider the prototypical normal means model in compound decision theory,

$$X_i \overset{ind}{\sim} N(\mu_i, \sigma^2) \tag{1}$$

for $1 \leq i \leq n$, where the means $\mu = (\mu_1, ..., \mu_n) \in \mathbb{R}^n$ and the noise level $\sigma > 0$ are parameters. As set out in the groundbreaking work of Robbins [47, 48, 67], the empirical Bayes setting is closely related,

$$\begin{aligned} \mu_i &\overset{iid}{\sim} G, \\ X_i \,|\, \mu_i &\overset{ind}{\sim} N(\mu_i, \sigma^2), \end{aligned} \tag{2}$$

where the prior $G$ is treated as an unknown parameter in contrast to a fully Bayesian approach. The empirical Bayes and compound decision theory literatures, motivated by the advent of scientific technologies (e.g. microarrays) enabling large-scale, parallel experiments, have witnessed tremendous methodological successes [18, 15, 67, 27, 19, 20, 44, 26, 24, 62, 65, 12, 21, 34, 55] and deep theoretical developments [66, 54, 23, 68, 31, 30, 46, 49, 33, 2, 8, 9, 45, 52].

In the Gaussian contexts of (1) and (2), existing theoretical work has largely focused on mean estimation or hypothesis testing when the variance $\sigma^2$ is known. The goal is to produce an estimator or a test and evaluate its performance with respect to the benchmark achieved by an oracle (termed oracle Bayes in [18, 31]) having access to information not available to the statistician (e.g. the prior $G$ in (2) or the empirical distribution $\frac{1}{n} \sum_{i=1}^n \delta_{\mu_i}$ in (1)). For estimation under square loss, the oracle estimator is $\mu_i^* = T^*(X_i)$ where

$$T^*(x) = E(\mu \,|\, X = x) = x + \sigma^2 \frac{f'(x)}{f(x)} \tag{3}$$

38th Conference on Neural Information Processing Systems (NeurIPS 2024).

is the posterior mean, under the prior $\mu \sim G$ in (2) or $\mu \sim \frac{1}{n}\sum_{i=1}^n \delta_{\mu_i}$ in (1), and is known as Tweedie's formula [16]. Here, $f$ denotes the density of the marginal distribution of $X$ and the second term in (3) is known as the Bayes correction. Tweedie's formula requires knowledge of the prior in order to compute $f$ and is thus available only to the oracle.

Broadly, there are two common approaches to mimicking the oracle [17]. Typically, the statistician either estimates the prior (known as $g$-modelling) or directly estimates the marginal distribution $f$ (known as $f$-modelling), and plugs in the resulting estimate into Tweedie's formula (3). Both approaches have been taken in the literature [66, 3, 2, 46, 28, 51], with the nonparametric maximum likelihood estimator (NPMLE) [33, 31, 34, 65, 51] being an especially popular method for $g$-modelling; regret bounds have been established under various assumptions on the class of priors.

Not much of the literature addresses the case of an unknown variance. The dearth of results is certainly not from want of motivation; estimation of the variance is necessary for mimicking the oracle estimator (3), construction of confidence intervals, hypothesis testing, and estimation of the signal-to-noise ratio among other statistical tasks. Rather, variance estimation has been recognized as a challenging problem in a wide range of high-dimensional and nonparametric models [61, 50, 35, 13, 11, 60, 29, 36, 4, 42]. A few articles have investigated a heteroskedastic version of (1) or (2) with unknown variances [40, 41, 25], but all of these articles essentially assume access to multiple i.i.d. observations per mean, thereby enabling direct estimation of each variance. The situation in (1) and (2) is quite different in that only one observation per mean is available; it is an honest sequence setting. Despite its clear importance, the fundamental limit of variance estimation in (1) and (2) has not been established in the literature.

This article investigates variance estimation from a minimax perspective. We will work in the compound decision setting (1), and so our results will also directly hold in the empirical Bayes setting (2). For the development of the minimax theory, the following parameter space will be considered,

$$\Theta(L) := \left\{ (\mu, \sigma) \in \mathbb{R}^n \times (0, \infty) : ||\mu||_\infty \le 1 \text{ and } \sigma \le L \right\}, \tag{4}$$

where $L > 0$. We will focus on the case where $L > 0$ is some large universal constant, and we will notationally suppress it by writing $\Theta$ to refer to (4). Furthermore, the choice of 1 in the bound $||\mu||_\infty \le 1$ is not essential; our results essentially go through when it is replaced by some other universal constant. The boundedness conditions are imposed to avoid triviality. Without boundedness, the minimax risk is easily seen to be infinite, $\inf_{\hat\sigma} \sup_{\mu \in \mathbb{R}^n, \sigma > 0} E\big(\big|\hat\sigma^2 - \sigma^2\big|^2\big) = \infty$, since the variance is not identifiable. In the empirical Bayes context (2), the constraint $||\mu||_\infty \le 1$ corresponds to the condition that $G$ is supported on $[-1, 1]$; the class of priors with bounded support is a popular choice for study in the empirical Bayes literature [46, 37, 3, 31, 38]. This article's goal is to obtain a sharp characterization of the minimax rate of variance estimation in (1) under square loss over the parameter space (4). Appendix E contains the notation used in this article.

## 1.1 Related work

The marginal distribution of the data in the empirical Bayes model (2) can be written as $X_1, ..., X_n \overset{iid}{\sim} G * N(0, \sigma^2)$ where $*$ denotes convolution. The problem of estimating $\sigma^2$ is a special case of variance estimation in a semiparametric convolution model [42, 4, 43], which encompasses generic noise distributions beyond Gaussian; the Fourier transform of the standardized (i.e. unit-scale) noise is assumed known. In their article [4], Butucea and Matias impose regularity conditions on $G$ by way of assumptions on its Fourier transform. Butucea and Matias point out it is essential $G$ is less smooth than the standardized noise distribution. In the Gaussian case, they assume the Fourier transform $\hat{G}$ does not vanish for large frequencies and its modulus does not decay faster than the Fourier transform of the standard Gaussian distribution $\omega \mapsto e^{-\omega^2/2}$. If $G$ may be somewhat smooth in the sense $|\hat{G}(\omega)| \ge ce^{-\alpha|\omega|^r}$ for $|\omega|$ sufficiently large where $0 < r < 2$, $\alpha > 0$, and $c > 0$ is an arbitrary constant, then Butucea and Matias construct a Fourier-based estimator which achieves $|\hat\sigma^2 - \sigma^2|^2 \lesssim (\log n)^{r-2}$ with high probability. In the rougher case $|\hat{G}(\omega)| \ge c|\omega|^{-\beta}$ for $|\omega|$ sufficiently large where $\beta > 1$, the faster rate $|\hat\sigma^2 - \sigma^2|^2 \lesssim (\log\log n/\log n)^2$ is achieved with high probability. Matching lower bounds are also obtained.

Though the results of [4] are sharp in their setting, the smoothness assumptions are unappealing and do not imply anything when $G$ is only assumed to have bounded support. In particular, $G$ being supported on $[-1, 1]$ does not imply $G$ must fall into one of the two cases considered by [4]. For

example, the Uniform$[-1, 1]$ distribution and the mixture $\frac{1}{2}\delta_{-1} + \frac{1}{2}\delta_1$ respectively have Fourier transforms $\omega \mapsto \sin(\omega)/\omega$ and $\omega \mapsto \cos(\omega)$, both of which have zeros that are arbitrarily large. Without the condition that $G$ has bounded support, the assumptions of [4] which stipulate $|\hat{G}(w)|$ is bounded away from zero for large $|\omega|$ are made to ensure identifiability of the variance. However, identifiability is automatically guaranteed when $G$ is assumed to be supported on $[-1, 1]$. Though the assumptions of [4] can be safely abandoned, it is not clear how, if it all, the approach of Butucea and Matias can be modified without smoothness conditions. Setting this serious issue aside, the keen reader might intuit from an uncertainty principle that if $G$ has a density $g$ which is not identically zero, then the Fourier transform of $g$ cannot decay too fast since it is compactly supported. In particular, it is well known it cannot decay faster than $Ce^{-c|\omega|}$, and so it might be guessed by taking $r = 1$ in the results of [4] that $\log^{-1} n$ may be optimal.

Moving away from smoothness conditions and assuming only that $G$ has bounded support, Matias [42] notices the moment generating function of $G * N(0, \sigma^2)$ is finite everywhere and is given by $M(\lambda) = \left(\int \exp(\lambda\mu)G(d\mu)\right)\exp(\lambda^2\sigma^2/2)$. Therefore, $\sigma^2$ is identifiable since $\lim_{\lambda \to \infty} \frac{2}{\lambda^2}\log M(\lambda) = \sigma^2$ due to the bounded support of $G$. With this observation in hand, Matias forms the empirical moment generating function $\hat{M}(\lambda) = \frac{1}{n}\sum_{i=1}^{n} e^{\lambda X_i}$, chooses $\lambda \asymp \sqrt{\log n}$, and defines the estimator $\hat{\sigma}^2 = \frac{2}{\lambda^2}\log\hat{M}(\lambda)$. Matias establishes the upper bound $E(|\hat{\sigma}^2 - \sigma^2|^2) \lesssim \frac{1}{\log n}$, but no matching lower bound is obtained. Though the use of the empirical moment generating function is clever, the simpler estimator $\hat{\sigma} = \max_{1 \leq i \leq n} X_i/\sqrt{2\log n}$ also achieves the same rate $|\hat{\sigma}^2 - \sigma^2|^2 \lesssim \frac{1}{\log n}$ with high probability as pointed out by [63].

A different perspective regards the marginal distribution $X_1, ..., X_n \overset{iid}{\sim} G * N(0, \sigma^2)$ as a Gaussian mixture model with mixing distribution $G$. Employing a moment-based approach, Wu and Yang [63] construct an estimator which achieves $|\hat{\sigma}^2 - \sigma^2|^2 \lesssim k^2 n^{-\frac{1}{k}}$ with high probability under the assumption $G$ is a $k$-atomic measure supported on $[-1, 1]$ with $k \lesssim \log n/\log\log n$. When $k$ is a fixed constant, a matching minimax lower bound is obtained. Their result highlights a rapid deterioration in the convergence rate as $k$ increases. In particular, in the generic case where $G$ need not be atomic (which can be intuited as $k \to \infty$), it might be expected the sharp rate will be logarithmic. Appendix A further discusses the results of [63].

Variance estimation has also been studied under sparsity or regularity assumptions on the means [11, 61, 5, 36]. In (1) under the assumption $||\mu||_0 \leq k < \frac{n}{2}$, an estimator achieving $|\hat{\sigma}^2 - \sigma^2| \lesssim \sigma^2\frac{k}{n}\log^{-1}\left(1 + k/\sqrt{n}\right)$ was constructed and shown to be minimax rate-optimal in [36] (see also [11, 7]). In the context of (2), faster rates of convergence were obtained by [5] when regularity on the means was assumed in addition to the sparsity. Fourier-based estimators were employed by all of [36, 11, 5]. Optimal estimation with Hölder-type regularity and without sparsity assumptions was obtained by way of kernel smoothing in [61]. These approaches critically exploit the regularity and/or sparsity, and all appear to fail when only boundedness $||\mu||_\infty \leq 1$ is assumed.

Beyond the sequence setting of (1) and (2), much recent attention has been directed at variance estimation in the linear regression model $Y_i = \langle X_i, \beta \rangle + \sigma Z_i$ where the noise $Z_i \sim N(0, 1)$ and the design $X_i \sim N(0, \Sigma)$ are independent. If $\Sigma \in \mathbb{R}^{p \times p}$ is known with bounded operator norm and $||\beta|| \lesssim 1$, then Dicker [14] showed the variance can be estimated at rate $|\hat{\sigma}^2 - \sigma^2|^2 \lesssim n^{-1} + pn^{-2}$ with high probability; notably, consistent estimation is possible even with $p \geq n$ without any sparsity assumptions on $\beta$. Assuming $\beta$ is $k$-sparse, [53] (see also [1, 22]) established the upper bound $n^{-1} + ((k\log p)/n)^2$. For particular regimes, optimality was established in [60], and the impossibility of consistent estimation for the dense regime was conjectured in the setting $n = o(p)$ and unknown $\Sigma$. Though some approaches were suggested in [13, 29], variance estimation with unknown $\Sigma$ was not settled until the impressive work of Kong and Valiant [35], who showed consistent estimation with $n = o(p)$ samples is surprisingly possible even when $k = p$. Assuming $||\beta|| \lesssim 1$ and $\Sigma$ has bounded condition number, an estimator can be constructed achieving, with high probability, error $|\hat{\sigma}^2 - \sigma^2| \leq \epsilon$ with sample complexity $n = O(\text{poly}(\log(1/\epsilon))p^{1 - \log^{-1}(1/\epsilon)})$. Without a condition number assumption on $\Sigma$, consistent estimation was shown to be possible with sample complexity $n = O(\text{poly}(1/\epsilon)p^{1 - \sqrt{\epsilon}})$. Moreover, Kong and Valiant [35] show these complexities are essentially tight. Their estimator is based on a clever polynomial approximation scheme. Though their article also covers certain covariate distributions which are not Gaussian, the sequence settings of (1) and (2) are not covered.

## 1.2 Main contribution

Our main result is a sharp characterization of the minimax rate of variance estimation over (4) in the compound decision setting (1),

$$\inf_{\hat{\sigma}} \sup_{(\mu,\sigma)\in\Theta} E_{\mu,\sigma}\left(\left|\hat{\sigma}^2 - \sigma^2\right|^2\right) \asymp \left(\frac{\log\log n}{\log n}\right)^2. \tag{5}$$

A few remarks are in order.

**Remark 1.** Though the slow, logarithmic nature of (5) is not surprising given the Gaussian mixture model results of [63] discussed earlier, it is notable a rate faster than $\log^{-1} n$ is achievable. The set of means $\mu$ in the parameter space (4) have no exploitable structure beyond $||\mu||_\infty \leq 1$, and so intuition suggests to examine the tails of the observations to estimate $\sigma^2$. This thinking may further suggest the least favorable prior might place substantial mass near $||\mu||_\infty \approx 1$, thus suggesting using an extreme order statistic.

Though natural, this intuition does not fully use the statistician's knowledge the noise is Gaussian. Conceptually, if the signal distribution looks very different from a Gaussian, then the signal might be easily disentangled from the noise, enabling easy variance estimation.[1] From a lower bound perspective, the least favorable prior for the means should resemble the noise. Indeed, our lower bound argument in Section 3 involves constructing a compactly supported prior which shares a growing number of moments with a particular Gaussian distribution. For technical reasons elaborated on in Section 3, it turns out one can only match $O(\log n / \log\log n)$ moments, which subsequently yields a lower bound of order $(\log\log n / \log n)^2$ for variance estimation.

The quantitative constraint on the number of moments which can be matched in the lower bound construction inspires hope it may be possible to estimate at a rate faster than $\log^{-1} n$, and it encourages developing some kind of method-of-moments estimator. Though such a strategy has seen success in a linear regression setting [35], complications seem to appear in the sequence model (1) as discussed in Remark 2. Cumulants, which are closely related to moments, have much more convenient properties, and so we develop a cumulant-based variance estimator which essentially requires estimating $O(\log n / \log\log n)$ cumulants, parallelling the lower bound.

**Remark 2.** Variance estimation in the sequence setting (1) turns out to be harder than in regression. Consider the linear regression model $Y_i = \langle X_i, \beta \rangle + \sigma Z_i$ where $\beta \in \mathbb{R}^p$ is unknown, and the noise $Z_i \sim N(0,1)$ is independent of the design $X_i \sim N(0,\Sigma)$. For comparison with the sequence model (1), consider the setting $p = n$ and $||\Sigma|| \vee ||\beta|| \lesssim 1$. Without any condition number assumption on $\Sigma$, Kong and Valiant [35] construct an estimator achieving, with high probability, the rate (translating their sample complexity result to estimation risk) $|\hat{\sigma}^2 - \sigma^2|^2 \lesssim (\log\log n / \log n)^4$. Though this result is not directly comparable to (5) since Kong and Valiant impose the $\ell_2$-norm constraint $||\beta|| \lesssim 1$ whereas the space (4) imposes an $\ell_\infty$-norm constraint more natural for compound decision theory, it is helpful for intuition to appreciate how the regression setting differs from the sequence setting to yield a "faster" rate.

In regression, it is clear the second marginal moment is $E(Y_1^2) = \langle \beta, \Sigma\beta \rangle + \sigma^2$, and so variance estimation is (up to parametric rate) equivalent to estimation of $\langle \beta, \Sigma\beta \rangle$. By changing focus to this target, $\sigma^2$ is now viewed as a nuisance. The key advantage in regression is that the covariates are a source of fresh randomness that is independent of the noise. Since $X_i$ is correlated with $Y_i$ but independent of $Z_i$, it is clear $E(Y_1 \langle X_1, X_2 \rangle Y_2) = \langle \beta, \Sigma^2\beta \rangle$, $E(Y_1 \langle X_1, X_2 \rangle \langle X_2, X_3 \rangle Y_3) = \langle \beta, \Sigma^3\beta \rangle$, and so on. In other words, unbiased estimators of the terms $\left\{\langle \beta, \Sigma^k\beta \rangle\right\}_{k=2}^n$ are easily constructed due to the availability of the random covariates $\{X_i\}_{i=1}^n$. The clever idea of Kong and Valiant [35] is to reach for polynomial approximation tools and approximate $\langle \beta, \Sigma\beta \rangle$ by the readily estimable $\left\{\langle \beta, \Sigma^k\beta \rangle\right\}_{k=2}^n$.

The sequence model (1) is equivalent to linear regression with a deterministic orthogonal design, which is quite different from the random design considered above. The target $||\mu||^2$ is the target analogous to $\langle \beta, \Sigma\beta \rangle$. The only accessible randomness in (1) are the responses themselves $\{X_i\}_{i=1}^n$. The marginal moments $\frac{1}{n}\sum_{i=1}^n E(X_i^k) = \frac{1}{n}\sum_{i=1}^n E((\mu_i + \sigma Z_i)^k)$ all involve the nuisance $\sigma$, thus precluding an application of Kong and Valiant's idea. The interesting phenomenon that design

---

[1]At a high-level, this is the core spirit of the results in the deconvolution literature (e.g. [4, 42]). However, we do not adopt a Fourier-based lens due to the issues discussed earlier.

properties can have substantial effects on minimax rates has been noted in the literature in other settings [11, 6, 59], and we suspect this is the cause for the slower rate (5) for variance estimation in the sequence model (1).

**Remark 3.** Given the central position of the Gaussian sequence models (1) and (2) in compound decision theory and empirical Bayes theory (as well as their status at the core of mathematical statistics more broadly), the cumulant-based variance estimator we propose in this article is designed to exploit the Gaussian character of the noise. It turns out incorporating noise information is, in some sense, essential; in Section 4 it is discussed that noise agnosticism implies the impossibility of consistent variance estimation. Interestingly, the results of Kong and Valiant [35] in the linear regression setting allow for an unknown noise distribution.[2] There appears to be subtle interplay between properties of the design and the noise; a careful study remains an open problem.

## 2   A cumulant-based estimator

As noted in Remark 1, the variance estimator we will propose is cumulant-based. Cumulants, though perhaps not as familiar, are related to the moments of a random variable. But, in contrast to moments, they have particularly nice attributes for the purpose of variance estimation in (1) and (2). Some key properties of cumulants are briefly reviewed before the estimation methodology is developed.

### 2.1   A brief review of cumulants

Suppose $Y$ is a random variable such that its moment generating function $M(\lambda) := E(e^{\lambda Y})$ exists for all $\lambda \in \mathbb{R}$. The cumulant generating function is defined by $K(\lambda) := \log M(\lambda)$, and it admits the power series $K(\lambda) = \sum_{r=1}^{\infty} \frac{\kappa_r \lambda^r}{r!}$ where $\kappa_r := K^{(r)}(0)$ is defined to be the $r$th cumulant. Though $\kappa_1$ and $\kappa_2$ coincide respectively with the mean and variance of $Y$, the higher cumulants do not have a simple relationship with the moments. However, the first $r$ cumulants are determined completely by the first $r$ moments; likewise, the first $r$ moments are determined completely by the first $r$ cumulants. In fact, an explicit correspondence is available via Bell polynomials.

**Definition 1.** *The incomplete Bell polynomial $B_{r,l}$ for $l \leq r$ is given by*

$$B_{r,l}(x_1, ..., x_{r-l+1}) = \sum \frac{r!}{j_1! j_2! \cdots j_{r-l+1}!} \left(\frac{x_1}{1!}\right)^{j_1} \left(\frac{x_2}{2!}\right)^{j_2} \cdots \left(\frac{x_{r-l+1}}{(r-l+1)!}\right)^{j_{r-l+1}}$$

*where the sum is taken over all sequences $j_1, j_2, j_3, ..., j_{r-l+1}$ of nonnegative integers such that $j_1 + ... + j_{r-l+1} = l$ and $j_1 + 2j_2 + 3j_3 + ... + (r-l+1)j_{r-l+1} = r$.*

The correspondence between moments and cumulants is given by

$$\kappa_r = \sum_{l=1}^{r} (-1)^{l-1} (l-1)! \, B_{r,l}(m_1, m_2, ..., m_{r-l+1}), \tag{6}$$

$$m_r = \sum_{l=1}^{r} B_{r,l}(\kappa_1, \kappa_2, \kappa_3, ..., \kappa_{r-l+1}), \tag{7}$$

where $m_r = E(Y^r)$ denotes the $r$th moment of $Y$. As is evident by its definition, the cumulant generating function plays nicely with convolutions. The cumulant generating function of the random variable $Y_1 + Y_2$ for independent $Y_1$ and $Y_2$ is given by the sum of the individual cumulant generating functions. Consequently, the $r$th cumulant of the sum is the sum of the $r$th cumulants.

### 2.2   Identifying the noise variance from the marginal cumulants

In the model (1), the second moment exhibits $\frac{1}{n} \sum_{i=1}^{n} X_i^2 = ||\mu||^2/n + \sigma^2 + O_P(n^{-1/2})$ due to the boundedness of $||\mu||_\infty$ and $\sigma$. It is clear estimation of $\sigma^2$ is equivalent to estimation of $||\mu||^2/n$ up to a (negligible) parametric slowdown in the rate. Associated with any estimator $\hat{Q}$ for the quadratic functional $\frac{||\mu||^2}{n}$ is a corresponding variance estimator $\hat{\sigma}^2 = \frac{1}{n} \sum_{i=1}^{n} X_i^2 - \hat{Q}$. Hence, it suffices

---

[2]Their results also allow for a quite general class of covariate distributions beyond centered multivariate Gaussians.

to focus attention on quadratic functional estimation. The estimation methodology is more plainly motivated in the empirical Bayes context (2), so the following discussion will develop ideas in that setting.

In the context of (2), the goal is to estimate the second moment of $G$. Let $\kappa_r$ and $m_r$ denote the $r$th cumulant and moment respectively of the marginal distribution $G * N(0, \sigma^2)$ of the data. Likewise, let $\gamma_r$ and $\nu_r$ denote the $r$th cumulant and moment respectively of $G$. Since the noise is mean zero and $\sigma$ is bounded, the mean of $G$ can be estimated at parametric rate, and so it is equivalent (up to a negligible parametric slowdown) to estimate the variance of $G$, i.e. $\gamma_2$.

It is not immediately clear how to identify $\gamma_2$ from the cumulants $\kappa_r$ of the marginal distribution of the data $G * N(0, \sigma^2)$. Consider the moment generating function of $N(0, \sigma^2)$ is $\lambda \mapsto \exp(\lambda^2 \sigma^2 / 2)$, and so all the cumulants (except the second) are equal to zero. Therefore, $\kappa_1 = \gamma_1, \kappa_2 = \gamma_2 + \sigma^2$, and $\kappa_r = \gamma_r$ for $r \geq 3$. The upshot is we are able to directly estimate all the cumulants of $G$, except the second, from $\{X_i\}_{i=1}^n$. Unfortunately, it is impossible to generically reconstruct the second cumulant from the other cumulants. Indeed, the collection of all centered Gaussians differ only in their second cumulant (the rest are zero).

It turns out it is possible to identify $\gamma_2$ by exploiting the boundedness of $G$'s support. For $r \geq 2$, define the function $M_r : [0, \infty) \to \mathbb{R}$ with

$$M_r(\gamma) = \sum_{l=1}^{r} B_{r,l}(\gamma_1, \gamma, \gamma_3, ..., \gamma_{r-l+1}) \tag{8}$$

where $B_{r,l}$ is an incomplete Bell polynomial (see Definition 1). Note by (7) that $M_r$ gives the $r$th moment associated to the cumulant sequence $(\gamma_1, \gamma, \gamma_3, \ldots)$ if it is a valid cumulant sequence. The following proposition asserts a variational representation of $\gamma_2$.

**Proposition 1.** *If $G$ is supported on $[-1, 1]$, then $\gamma_2 = \sup\{\gamma \in [0, 1] : |M_r(\gamma)| \leq 1 \text{ for all } r \geq 2\}$.*

*Proof.* For ease of notation, let $\gamma^* = \sup\{\gamma \in [0, 1] : |M_r(\gamma)| \leq 1 \text{ for all } r \geq 2\}$. It is clear $\gamma_2 \leq \gamma^*$ since $M_r(\gamma_2) = \nu_r \in [-1, 1]$ for all $r \geq 2$. To show the lower bound, fix $\delta > 0$. For any $\tau^2 \geq \delta$, observe that the cumulant sequence of the distribution $G * N(0, \tau^2)$ is $(\gamma_1, \gamma_2 + \tau^2, \gamma_3, \gamma_4, ...)$. Since $G * N(0, \tau^2)$ does not have all moments contained in $[-1, 1]$, it follows $\gamma_2 + \tau^2$ is not a feasible point. Since this holds for all $\tau^2 \geq \delta$, it immediately follows $\gamma^* \leq \gamma_2 + \delta$. Since $\delta > 0$ was arbitrary, we have shown $\gamma^* \leq \gamma_2$. $\square$

Proposition 1 enables recovery of $\gamma_2$ from the other cumulants. However, it requires certifying *all* putative moments $\{M_r(\gamma)\}_{r=2}^{\infty}$ live in $[-1, 1]$, which is a difficult task given only a finite amount of data. One idea is to approximate $\gamma_2$ by certifying only that an $r$th putative moment lies in $[-1, 1]$ for some large choice of $r$. Specifically, define for $r \geq 2$,

$$\tilde{\gamma}_2(r) := \sup\{\gamma \in [0, 1] : |M_r(\gamma)| \leq 1\}. \tag{9}$$

**Proposition 2.** *If $r$ is even, then $|\tilde{\gamma}_2(r) - \gamma_2| \leq \frac{C}{r}$ for some universal constant $C > 0$.*

*Proof.* By (9) and Proposition 1, it follows $\gamma_2 \leq \tilde{\gamma}_2(r)$. Therefore, $M_r(\tilde{\gamma}_2(r))$ is the $r$th moment of $G * N(0, \tilde{\gamma}_2(r) - \gamma_2)$. Since $r$ is even, taking $\mu \sim G$ and $Z \sim N(0, 1)$, it follows $1 \geq |M_r(\tilde{\gamma}_2(r))| = E((\mu + \sqrt{\tilde{\gamma}_2(r) - \gamma_2}Z)^r) = \sum_{l=0}^{r} \binom{r}{l} E(\mu^{r-l})(\tilde{\gamma}_2(r) - \gamma_2)^{l/2} E(Z^l)$. If $r - l$ is odd, then $l$ must be odd since $r$ is even, and so we must have $E(Z^l) = 0$. It thus follows $1 \geq \sum_{l \leq r, \, l \text{ even}} \binom{r}{l} E(|\mu|^{r-l})(\tilde{\gamma}_2(r) - \gamma_2)^{l/2} E(Z^l) \geq (\tilde{\gamma}_2(r) - \gamma_2)^{r/2} E(Z^r)$. Hence, $|\tilde{\gamma}_2(r) - \gamma_2| \leq E(Z^r)^{-2/r} = \pi^{-1/r} \left(2^{r/2} \Gamma\left(\frac{r+1}{2}\right)\right)^{-2/r}$. It follows from Stirling's approximation that for some small universal constant $c > 0$ whose value may change from instance to instance, we have $\Gamma\left(\frac{r+1}{2}\right)^{2/r} \geq c \left(\sqrt{\pi(r-1)} \left(\frac{r-1}{2e}\right)^{\frac{r-1}{2}}\right)^{2/r} \geq cr$. This immediately yields the claimed bound. The proof is complete. $\square$

The methodological strategy is in place; an estimator will be constructed to estimate $\tilde{\gamma}_2(r)$ for a well-chosen value of $r$ to balance the estimator's variance with the bias from Proposition 2.

## 2.3  Methodology

We return to the compound setting of (1); there is now no "ground truth" prior $G$. However, a recurring theme of compound decision theory [48, 47] is that it mimics the empirical Bayes theory as if the prior were the empirical distribution of the means. Note the quadratic functional of interest $||\mu||^2/n$ is precisely equal to the second moment of the empirical distribution of the means.

To describe our estimation strategy, some preliminary development is necessary. For even $r$, set $m_r$ to be the $r$th moment of the distribution $(\frac{1}{n}\sum_{i=1}^n \delta_{\mu_i}) * N(0, \sigma^2)$. For odd $r$, set $m_r = 0$. Observe $m_1, m_2, ...$ are the moments of the symmetrized distribution $(\frac{1}{n}\sum_{i=1}^n (\frac{1}{2}\delta_{\mu_i} + \frac{1}{2}\delta_{-\mu_i})) * N(0, \sigma^2)$. Thus, we are able to conceptually place ourselves in the context of Section 2.2 by making the choice of "prior"[3] $G = \frac{1}{n}\sum_{i=1}^n (\frac{1}{2}\delta_{\mu_i} + \frac{1}{2}\delta_{-\mu_i})$. Let us adopt the notation of Section 2.2, namely let $\kappa_r$ denote the $r$th cumulant of $G * N(0, \sigma^2)$, and let $\gamma_r$ and $\nu_r$ denote the $r$th cumulant and moment respectively of $G$. As noted in Section 2.2, we have $\kappa_r = \gamma_r$ for $r \neq 2$. Furthermore, note the quadratic functional of interest $||\mu||^2/n$ is exactly $\gamma_2$.

The strategy is to estimate $\tilde{\gamma}_2(r)$ given by (9) for a carefully chosen value of $r$. In pursuit of this strategy, estimators for the cumulants (except the second) will now be constructed. Define the moment estimators

$$\hat{m}_r := \begin{cases} \frac{1}{n}\sum_{i=1}^n X_i^r & \text{if } r \text{ is even,} \\ 0 & \text{otherwise.} \end{cases} \tag{10}$$

Cumulant estimators are obtained via plugging in to (6),

$$\hat{\gamma}_r := \sum_{l=1}^r (-1)^{l-1}(l-1)! \, B_{r,l}(\hat{m}_1, \hat{m}_2, ..., \hat{m}_{r-l+1}) \tag{11}$$

for $r \neq 2$. To estimate $\tilde{\gamma}_2(r)$, an estimator of the function $M_r$ given by (8) is needed. Define the (random) function $\hat{M}_r : [0, \infty) \to \mathbb{R}$ given by

$$\hat{M}_r(\gamma) = \sum_{l=1}^r B_{r,l}(\hat{\gamma}_1, \gamma, \hat{\gamma}_3, ..., \hat{\gamma}_{r-l+1}). \tag{12}$$

For $r \geq 2$ and $\varepsilon > 0$, define the estimator

$$\hat{\gamma}_2(r) := \sup\left\{\gamma \in [0, 1] : \left|\hat{M}_r(\gamma)\right| \leq 1 + \varepsilon\right\} \tag{13}$$

where $r, \varepsilon$ are tuning parameters to be chosen. The high-level justification behind $\hat{\gamma}_2(r)$ lies in the approximation,

$$M_r(\hat{\gamma}_2(r)) \approx \hat{M}_r(\hat{\gamma}_2(r)) \approx M_r(\tilde{\gamma}_2(r)). \tag{14}$$

Suppose it could be shown $\hat{M}_r(\gamma)$ concentrates around $M_r(\gamma)$ uniformly over $\gamma \in [0, 1]$. If so, then the first approximation in (14) follows. The Bell polynomial structure is quite convenient as it facilitates a straightforward proof of the desired uniform concentration (see Appendix B.2); cumbersome empirical process theory is avoided. Since both $\hat{\gamma}_2(r)$ and $\tilde{\gamma}_2(r)$ are defined in terms of supremums in (13) and (9), both $\hat{M}_r(\hat{\gamma}_2(r))$ and $M_r(\tilde{\gamma}_2(r))$ will be near one, intuitively yielding the second approximation in (14). Thus $|M_r(\hat{\gamma}_2(r)) - M_r(\tilde{\gamma}_2(r))|$ is small, and since $M_r$ is an $\frac{r}{2}$-degree polynomial[4], intuition suggests $|\hat{\gamma}_2(r) - \tilde{\gamma}_2(r)|$ can be controlled via an $\frac{r}{2}$-degree Taylor expansion. Combining with the approximation error bound provided by Proposition 2 delivers a bound on $|\hat{\gamma}_2(r) - \gamma_2|$, as the following result states.

**Proposition 3.** *Fix $\varepsilon > 0$ and even $r$. Let $\mathcal{E} = \left\{\sup_{\gamma \in [0,1]} \left|\hat{M}_r(\gamma) - M_r(\gamma)\right| \leq \varepsilon\right\}$ where $\hat{M}_r$ and $M_r$ are given by (12) and (8) respectively. On the event $\mathcal{E}$, we have $|\hat{\gamma}_2(r) - \gamma_2| \leq ((r/2)! \cdot 2\varepsilon)^{2/r} + \frac{C}{r}$ where $\hat{\gamma}_2(r)$ is given by (13) and $C > 0$ is a universal constant.*

---

[3]The reader should understand this choice of "prior" as essentially the same as the usual compound decision-theoretic choice of the empirical distribution of the means. However, it happens to be more convenient to take the symmetrized version.

[4]Note that in the Bell polynomial $B_{r,l}(\gamma_1, \gamma, \gamma_3, ..., \gamma_{r-l+1})$, it follows by Definition 1 that the power of $\gamma$ is given by $j_2$ which must satisfy $2j_2 \leq r$.

*Proof.* It follows from the definition (9) of $\tilde{\gamma}_2(r)$ and Proposition 1 that $\gamma_2 \leq \tilde{\gamma}_2(r)$. On the event $\mathcal{E}$, it follows from the definition of $\hat{\gamma}_2(r)$ and $|M_r(\tilde{\gamma}_2(r))| \leq 1$ that $\tilde{\gamma}_2(r) \leq \hat{\gamma}_2(r)$. Therefore, on the event $\mathcal{E}$, it follows by Lemmas 1 and 2 that $1 = M_r(\tilde{\gamma}_2(r)) \leq M_r(\hat{\gamma}_2(r)) \leq \hat{M}_r(\hat{\gamma}_2(r)) + \varepsilon \leq 1 + 2\varepsilon$. This implies $0 \leq M_r(\hat{\gamma}_2(r)) - M_r(\tilde{\gamma}_2(r)) \leq 2\varepsilon$, and so an application of Lemma 3 yields $0 \leq \hat{\gamma}_2(r) - \tilde{\gamma}_2(r) \leq ((r/2)! \cdot 2\varepsilon)^{2/r}$. Combining this bound with the approximation error bound of Proposition 2 delivers the desired result. $\qquad\square$

To obtain an estimation error bound, it remains to investigate the stochastic error, i.e. the typical magnitude of $\sup_{\gamma \in [0,1]} |\hat{M}_r(\gamma) - M_r(\gamma)|$, so that the first approximation in (14) is justified. The cumulant estimators $\{\hat{\gamma}_l\}_{1 \leq l \leq r, l \neq 2}$ used to define $\hat{M}_r(\gamma)$ are constructed by plugging in sample moment estimators for the even moments. Heuristically, only a slow concentration rate can be expected because the number of moments $r$ being estimated is large.

**Proposition 4.** *Suppose $(\mu, \sigma) \in \Theta$ and $c^*, \beta > 0$ are universal constants. If $\delta \geq \frac{1}{\log^\beta n}$, then there exist universal constants $C_{\beta,*}, C, C' \geq 1$ depending only on $\beta$ and $c^*$ such that the following holds. If $2 \leq r \leq \frac{1}{C_{\beta,*}} \frac{\log n}{\log \log n}$ and $\sigma^2 \geq \frac{c^*}{r}$, then $P_{\mu,\sigma} \left\{ \sup_{\gamma \in [0,1]} \left| \hat{M}_r(\gamma) - M_r(\gamma) \right| \leq \frac{Ce^{C'r \log r}}{\sqrt{n\delta}} \right\} \geq 1 - \delta$.*

With the stochastic error addressed by Proposition 4, the ingredients are in place to furnish a risk bound for the variance estimator associated to (13). There is the slight technical point that Proposition 4 applies only when $\sigma^2 \gtrsim \frac{1}{r}$. This condition may be an artifact of the proof, and the conclusion of Proposition 4 might continue to hold for small $\sigma$. However, Proposition 4 turns out to suffice for the purpose of variance estimation as we can incorporate a truncation step. The variance estimator is defined as follows. For an even integer $r$ and a real $\varepsilon > 0$, define

$$\hat{\sigma}^2 = \left( \frac{1}{n} \sum_{i=1}^n X_i^2 - \hat{\gamma}_2(r) \right) \mathbb{1}_{\left\{ \max_{1 \leq i \leq n} X_i > 4\sqrt{\frac{\log n}{r}} \right\}} \tag{15}$$

where $\hat{\gamma}_2(r)$ is given by (13).

**Theorem 1.** *Let $\delta = \frac{1}{\log^2 n}$. Let $C_{\beta,*}, C,$ and $C'$ be the universal constants from Proposition 4 corresponding to $c^* = 1$ and $\beta = 2$. There exists a universal constant $C^* \geq C_{\beta,*}$ such that the following holds. If $r$ is the largest even integer less than or equal to $\frac{1}{C^*} \frac{\log n}{\log \log n}$ and $\varepsilon = \frac{Ce^{C'r \log r}}{\sqrt{n\delta}}$, then*

$$\sup_{(\mu,\sigma) \in \Theta} E_{\mu,\sigma} \left( |\hat{\sigma}^2 - \sigma^2|^2 \right) \lesssim \left( \frac{\log \log n}{\log n} \right)^2,$$

*where $\hat{\sigma}^2$ is given by (15).*

Since $||\mu||_\infty \leq 1$ implies that $\max_{1 \leq i \leq n} X_i$ is typically no larger than $1 + \sqrt{2\sigma^2 \log n}$, the truncation choice[5] $4\sqrt{\frac{\log n}{r}}$ essentially results in using the trivial estimator $\hat{\sigma}^2 = 0$ when $\sigma^2 \lesssim \frac{1}{r}$. On the other hand when $\sigma^2 \gtrsim \frac{1}{r}$, the stochastic error result of Proposition 4 is meaningful and it can be shown $\frac{1}{n} \sum_{i=1}^n X_i^2 - \hat{\gamma}_2(r)$ estimates $\sigma^2$ well.

## 3 Lower bound

The slow convergence rate of the cumulant-based estimator of Section 2 turns out to be the sharp rate.

**Theorem 2.** *There exist universal constants $C, c > 0$ such that $\inf_{\hat{\sigma}} \sup_{(\mu,\sigma) \in \Theta(2)} P_{\mu,\sigma} \{ |\hat{\sigma}^2 - \sigma^2|^2 \geq C (\frac{\log \log n}{\log n})^2 \} \geq c$.*

The lower bound is proved through a moment matching technique employed in Le Cam's two point method. The intuition for the construction follows from appreciating the constraint $||\mu||_\infty \leq 1$. As discussed, the boundedness constraint is imposed to ensure $\sigma^2$ is identifiable. Though quite

---

[5]The choice of 4 is not crucial and could be replaced by some other constant with no change to the conclusion of Theorem 1.

elementary, it is conceptually useful for our construction to examine why $\sigma^2$ is unidentifiable if no constraints are imposed. Specifically, it can be understood from a lower bound perspective by examining the reduction to the following Bayesian testing problem, $H_0 : \mu = 0$ and $\sigma^2 = 1 + \tau^2$ against $H_1 : \mu_i \sim N(0, \tau^2)$ and $\sigma^2 = 1$. Observe that under both hypotheses the data are $X_1, ..., X_n$ are i.i.d. draws from $N(0, 1 + \tau^2)$. The hypotheses are indistinguishable yet the variances differ by $\tau^2$ under $H_0$ and $H_1$. Therefore, any variance estimator incurs square loss of at least $\tau^4$; taking $\tau \to \infty$ establishes infinite estimation risk is inescapable.

Under the boundedness constraint $||\mu||_\infty \leq 1$, the choice of prior $\mu_1, ..., \mu_n \overset{iid}{\sim} N(0, \tau^2)$ is no longer feasible. One idea is to find a distribution $G$ supported on $[-1, 1]$ such that the marginal data distribution $G * N(0, 1)$ is indistinguishable from $N(0, \tau^2) * N(0, 1)$. This can be achieved by constructing $G$ to share a large number of moments with $N(0, \tau^2)$. To elaborate, if the first $2k-1$ moments match and $\tau < 1$, then (see Theorem 3.3.3 of [64]) we have $\chi^2(G * N(0, 1) \,||\, N(0, \tau^2) * N(0, 1)) \leq \frac{16}{\sqrt{2k-1}} \frac{\tau^{4k}}{1 - \tau^2}$. A well-known sufficient condition (e.g. see [58]) for the desired indistinguishability is that the $\chi^2$-divergence is at most $O(1/n)$. Hence, we seek a $G$ through moment matching.

How many moments can be matched under the constraint that $G$ be supported on $[-1, 1]$? Since the odd moments of a centered Gaussian are all zero and taking $G$ symmetric would thus match the odd moments, only the even moments require consideration. For $k$ even, the $k$th moment of $N(0, \tau^2)$ is at least, by Stirling's approximation, $c\tau^k e^{k \log k / 2}$ for some small constant $c > 0$. On the other hand, the $k$th moment of $G$ must lie in $[-1, 1]$. Therefore, a necessary condition for the $k$th moments to match is $\tau^k e^{k \log k / 2} \lesssim 1$. This crude reasoning already shows $G$ can only match at most $O(1/\tau^2)$ moments.

The actual construction of $G$ relies on the technology of Gaussian quadrature. With Proposition 9 and Lemma 8, it can be shown that the $k$-atomic Gaussian quadrature $G = \sum_{i=1}^{k} w_i \delta_{\tau z_i}$, where $\{z_i\}_{i=1}^{k}$ are the roots of the $k$th Hermite polynomial, is supported on the interval $[-\sqrt{\tau^2(4k-4)}, \sqrt{\tau^2(4k-4)}]$ and shares the first $2k - 1$ moments with $N(0, \tau^2)$. A sufficient condition for $G$ to be supported on $[-1, 1]$ is $k \leq 1/(4\tau^2)$. Therefore, a properly supported $G$ which matches $\Omega\left(1/\tau^2\right)$ moments of $N(0, \tau^2)$ can be constructed, validating that our crude reasoning from earlier is tight. The problem now boils down to choosing $\tau^2$. Since $k \asymp \tau^{-2}$, it follows from the $\chi^2$-divergence bound implied by moment matching that choosing $\tau^2 \asymp \frac{\log \log n}{\log n}$ yields the desired $\frac{1}{n}$ bound on the $\chi^2$-divergence and is precisely the desired separation.

# 4 Noise agnosticism

The cumulant-based variance estimator proposed in this paper heavily relies on the Gaussian character of the noise, specifically that all cumulants (except the first- and second-order) of a Gaussian are equal to zero regardless of the Gaussian's mean and variance parameters. From a minimax perspective, it turns out it is a fundamental necessity to exploit information about the noise. Concretely, it can be shown that consistent variance estimation is impossible if nothing beyond subgaussianity (see Definition 2) is assumed. The impossibility can be seen by appealing to Le Cam's two-point method to reduce the problem to a two-point testing problem and using the following simple construction,

$$H_0 : \mu_i \overset{iid}{\sim} \text{Rademacher}\,(1/2)\,, \sigma^2 = 1,\ \text{and}\ \xi_i \overset{iid}{\sim} \text{Rademacher}\,(1/2)\,,$$

$$H_1 : \mu = 0, \sigma^2 = 2,\ \text{and}\ \xi_i \sqrt{2} \overset{iid}{\sim} \text{Rademacher}\,(1/2) * \text{Rademacher}\,(1/2)\,.$$

It is clear both noise distributions are 2-subgaussian. Furthermore, the data $\{X_i\}_{i=1}^{n}$ are independent and identically distributed according to $\text{Rademacher}\,(1/2) * \text{Rademacher}\,(1/2)$ under both hypotheses, and so it is impossible to distinguish $H_0$ and $H_1$. Since the separation between the choices of $\sigma^2$ between the two hypotheses is 1, it follows consistent estimation is impossible. This is formally stated in the following proposition without proof.

**Proposition 5.** *For $a > 0$, let $\Xi_a := \{P : P \text{ is } a\text{-subgaussian, has mean } 0 \text{ and variance } 1\}$. Let $P_{\mu, \sigma, P_\xi}$ denote the joint distribution of the observations from the sequence model, $X_i = \mu_i + \sigma \xi_i$ for $1 \leq i \leq n$ where $\xi_i \sim P_\xi$ are i.i.d. Then $\inf_{\hat{\sigma}} \sup_{(\mu, \sigma) \in \Theta(2), P_\xi \in \Xi_2} E_{\mu, \sigma, P_\xi} \left(|\hat{\sigma}^2 - \sigma^2|^2\right) \gtrsim 1$.*

In this sense it is necessary to exploit finer information about the noise, and so our cumulant-based estimator is designed to exploit the noise's Gaussian character in (1).

## Acknowledgments and Disclosure of Funding

This work was supported in part by NSF Grant ECCS-2216912. The author thanks Chao Gao for helpful discussions.

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

# Appendices to "Variance estimation in compound decision theory under boundedness"

The appendices are organized as follows. Appendix A discusses the connection to Gaussian mixture model results of [63]. The remaining proofs for the results of Section 2.3 are presented in Appendix B and the proof of Theorem 2 is presented in Appendix C. Appendix D contains auxiliary definitions and results. Finally, Appendix E describes the notation used in the main text and the appendices.

## A   Relationship to Gaussian mixture model

Viewing the marginal distribution $X_1, ..., X_n \overset{iid}{\sim} G * N(0, \sigma^2)$ as a Gaussian mixture model, Wu and Yang [63] assume $G$ is a $k$-atomic distribution with bounded support and, for $k \lesssim \frac{\log n}{\log \log n}$, show that the variance estimator furnished by Lindsay's algorithm [39] achieves $|\hat{\sigma}^2 - \sigma^2|^2 \lesssim k^2 n^{-\frac{1}{k}}$ with high probability. A minimax lower bound of order $n^{-\frac{1}{k}}$ is obtained, establishing optimality for fixed $k$.

The main focus of Wu and Yang's article is estimation of $G$ with respect to the Wasserstein 1-distance. In the case where the variance $\sigma^2$ is known, they develop a *denoised method of moments* (DMM) estimator and prove it achieves the minimax estimation rate. Notably, their results cover the case where $G$ has a continuous density (see Theorem 5 in [63]); in this case, $G$ is approximated by a finite mixture with $c\frac{\log n}{\log \log n}$ atoms (for some small constant $c > 0$), and the approximation is estimated by the DMM estimator.

When $\sigma^2$ is unknown, Wu and Yang study Lindsay's algorithm (Algorithm 3 in [63]). However, their article [63] only addresses the case $k \leq c\frac{\log n}{\log \log n}$ and no analogue of Theorem 5 is offered; no error bound for the variance estimator is offered either. For continuous $G$, a natural idea is to mimic the known variance case by approximating the continuous $G$ by a $\tilde{k}$-atomic measure and running Lindsay's algorithm with this choice $\tilde{k}$. One may hope the fact that the data actually come from the continuous $G$ and not the $\tilde{k}$-atomic approximation does not pose a problem.

Unfortunately, it appears the analysis in [63] of Lindsay's algorithm strongly relies on the assumption that the true data-generating distribution $G$ is an atomic measure. To illustrate, let us examine the proof on page 1997 in [63]. Denote $\hat{\pi} = \hat{G} * N(0, \hat{\sigma}^2)$ and $\pi = G * N(0, \sigma^2)$ where $\hat{G}, \hat{\sigma}^2$ are the outputs of Lindsay's algorithm. Consider the case $\sigma \leq \hat{\sigma}$, and denote $G' = \hat{G} * N(0, \hat{\sigma}^2 - \sigma^2)$. Following their proof, though Proposition 3 in [63] cannot now be directly invoked since $G$ is not $\tilde{k}$-atomic, one may hope a modification of Proposition 3's proof might be possible. However, Proposition 3 relies on Lemma 13 (found in the supplementary material of [63]), which we would like to apply to the measures $\hat{G} * N(0, \hat{\sigma}^2 - \sigma^2)$ and $G$. Though Lemma 13 does not require $\hat{G}$ to be an atomic distribution, it does require $G$ to be $\tilde{k}$-atomic and the proof of Lemma 13 makes essential use of this assumption. It is not clear whether this can be circumvented in a straightforward manner. It is an interesting open problem to obtain an analogue of Theorem 5 in the case of unknown variance, and also to establish whether or not Lindsay's algorithm can achieve the optimal estimation rate.

## B   Upper bound

The high-level justification behind the development of the estimation methodology presented in Section 2.3 is the approximation (14). As discussed, the two key pieces are the handling of the stochastic error $\sup_{\gamma \in [0,1]} |\hat{M}_r(\gamma) - M_r(\gamma)|$ and the approximation error $|\hat{\gamma}_2(r) - \tilde{\gamma}_2(r)|$, the latter of which is controlled through a Taylor expansion of $M_r$. Analytic properties of $M_r$ are presented in Appendix B.1 and the concentration of $\hat{M}_r$ is presented in Appendix B.2

### B.1   Analytic properties of $M_r$

The following results pertain to the function $M_r$ given by (8) and are made with the context of Section 2.2 in force.

**Lemma 1.** *If $r$ is even, then the function $M_r$ given by (8) is strictly increasing on the interval $(\gamma_2, \infty)$.*

*Proof.* Observe for $\gamma > \gamma_2$ that $M_r(\gamma)$ gives the $r$th moment of the distribution $G * N(0, \gamma - \gamma_2)$. In other words, letting $\mu \sim G$ and $Z \sim N(0, 1)$, we have

$$M_r(\gamma) = E\left(\left(\mu + \sqrt{\gamma - \gamma_2} Z\right)^r\right) = \sum_{l=0}^r \binom{r}{l} E(\mu^{r-l})(\gamma - \gamma_2)^{l/2} E(Z^l).$$

Consider that if $r - l$ is odd, then $l$ must also be odd since $r$ is even, which implies $E(Z^l) = 0$. Therefore, we can write

$$M_r(\gamma) = \sum_{\substack{0 \le l \le r, \\ l \text{ even}}} \binom{r}{l} E(|\mu|^{r-l})(\gamma - \gamma_2)^{l/2} E(Z^l).$$

Hence, $M_r$ is strictly increasing in $\gamma$ on the interval $(\gamma_2, \infty)$ as claimed. □

**Lemma 2.** *If $r$ is even, then $M_r(\tilde{\gamma}_2(r)) = 1$ where $M_r$ is given by (8) and $\tilde{\gamma}_2(r)$ is given by (9).*

*Proof.* Consider $M_r$ is a continuous function as it is a polynomial. Observe $M_r(\gamma_2) \le 1$ by Proposition 1. Furthermore, consider since $M_r(\gamma)$ is the $r$th moment of $G * N(0, \gamma - \gamma_2)$ for $\gamma > \gamma_2$ it follows $\lim_{\gamma \to \infty} M_r(\gamma) = \infty$. Furthermore, since Lemma 1 asserts $M_r$ is monotonically increasing on $(\gamma_2, \infty)$ and since $\tilde{\gamma}_2(r) \ge \gamma_2$, it follows by continuity that $M_r(\tilde{\gamma}_2(r)) = 1$. □

**Lemma 3.** *If $x \ge y \ge \gamma_2$ and $r$ is even, then $x - y \le ((r/2)! \, (M_r(x) - M_r(y)))^{2/r}$ where $M_r$ is given by (8).*

*Proof.* From the proof of Lemma 1 and taking $\mu \sim G$ and $Z \sim N(0, 1)$, we have for $\gamma \ge \gamma_2$,

$$M_r(\gamma) = \sum_{\substack{0 \le l \le r, \\ l \text{ even}}} \binom{r}{l} E(|\mu|^{r-l}) E(Z^l)(\gamma - \gamma_2)^{l/2}.$$

For $0 \le k \le \frac{r}{2}$ and for $\gamma \ge \gamma_2$, it follows from even $l$ that

$$M_r^{(k)}(\gamma) = \sum_{\substack{0 \le l \le r, \\ l \text{ even}, \\ l/2 \ge k}} \binom{r}{l} E(|\mu|^{r-l}) E(Z^l) \frac{(l/2)!}{(l/2 - k)!}(\gamma - \gamma_2)^{l/2-k} \ge E(|\mu|^{r-2k}).$$

Since $M_r(\gamma)$ is an $\frac{r}{2}$-degree polynomial in $\gamma$, Taylor expansion along with the above bound yields

$$M_r(x) - M_r(y) = \sum_{k=1}^{r/2} M_r^{(k)}(y) \frac{(x - y)^k}{k!} \ge \frac{(x - y)^{r/2}}{(r/2)!}.$$

Here, we have used that every term in the sum is nonnegative. In particular, $M_r^{(k)}(y) \ge 0$ and $(x - y)^k \ge 0$ since $x \ge y$. Rearranging gives the desired result. □

## B.2 Concentration of $\hat{M}_r$

As the definition of $\hat{M}_r$ in (12) relies on moment estimators $\hat{m}_r$ given by (10) and cumulant estimators $\hat{\gamma}_r$ given by (11), we will first collect results for these intermediate estimators. Further, recall from Section 2.3 that $m_r = \frac{1}{n} \sum_{i=1}^n E(X_i^r)$ if $r$ is even and $m_r = 0$ otherwise.

**Lemma 4.** *If $||\mu||_\infty \le 1$, then $\mathrm{Var}(\hat{m}_r) \le \frac{4^r + \sigma^{2r}(4r)^{2r}}{n}$.*

*Proof.* The claim is trivially true for odd $r$, so consider even $r$. Write $X_i = \mu_i + \sigma Z_i$ where $Z_i \overset{iid}{\sim} N(0,1)$. Observe by independence and $|\mu_i| \leq 1$, we have $\mathrm{Var}(\hat{m}_r) = \mathrm{Var}\left(\frac{1}{n}\sum_{i=1}^n X_i^r\right) = \frac{1}{n^2}\sum_{i=1}^n \mathrm{Var}(X_i^r) \leq \frac{1}{n^2}\sum_{i=1}^n E(X_i^{2r})$. By Jensen's inequality, we have $X_i^{2r} = 2^{2r}\left(\frac{\mu_i}{2} + \frac{\sigma Z_i}{2}\right)^{2r} \leq 2^{2r-1}\left(\mu_i^{2r} + (\sigma Z_i)^{2r}\right)$. Taking expectation yields

$$E(X_i^{2r}) \leq 2^{2r-1}(\mu_i^{2r} + \sigma^{2r}(2r-1)!!) \leq 2^{2r-1}\left(1 + \sigma^{2r}(2r-1)^{2r-1}\right) \leq 4^r + \sigma^{2r}(4r)^{2r},$$

which yields the desired result. $\qquad\square$

**Corollary 1.** *If $\|\mu\|_\infty \leq 1$ and $u > 0$, then $P_{\mu,\sigma}\left\{\max_{1 \leq l \leq r}|\hat{m}_l - m_l| > u\right\} \leq \frac{r4^r + r(1 \vee \sigma^2)^r(4r)^{2r}}{u^2 n}$.*

*Proof.* The result follows by union bound and Chebyshev's inequality with Lemma 4. $\qquad\square$

**Proposition 6.** *Suppose $(\mu, \sigma) \in \Theta(L)$ where $\Theta(L)$ is given by (4) and where $L > 0$ is a universal constant. There exists a large universal constant $C > 0$ such that the following holds. If $u > 0$ such that $\frac{C^r u}{1 \wedge \sigma^r} \in (0,1)$, then on the event $\{\max_{1 \leq l \leq r}|\hat{m}_l - m_l| \leq u\}$ we have*

$$\max_{\substack{1 \leq l \leq r, \\ l \neq 2}}|\hat{\gamma}_l - \gamma_l| \leq (1 + L^r r!) \cdot (r!)^2 \cdot (2r)^r \cdot \left|e^{\frac{C^r}{1 \wedge \sigma^r} r u} - 1\right|.$$

*Proof.* The argument borrows heavily from the proof of Lemma A.4 in [10]. For $1 \leq k \leq r$ with $k \neq 2$, we have by Definition 1,

$$|\hat{\gamma}_k - \gamma_k|$$

$$\leq \sum_{l=1}^k (l-1)!|B_{k,l}(\hat{m}_1, ..., \hat{m}_{k-l+1}) - B_{k,l}(m_1, ..., m_{k-l+1})|$$

$$\leq \sum_{l=1}^k (l-1)! \sum \frac{k!}{j_1! j_2! \cdots j_{k-l+1}!} \cdot$$

$$\left|\left(\frac{\hat{m}_1}{1!}\right)^{j_1}\left(\frac{\hat{m}_2}{2!}\right)^{j_2}\cdots\left(\frac{\hat{m}_{k-l+1}}{(k-l+1)!}\right)^{j_{k-l+1}} - \left(\frac{m_1}{1!}\right)^{j_1}\left(\frac{m_2}{2!}\right)^{j_2}\cdots\left(\frac{m_{k-l+1}}{(k-l+1)!}\right)^{j_{k-l+1}}\right|$$

where the sum is taken over all sequences $j_1, j_2, ..., j_{k-l+1}$ of nonnegative integers such that $j_1 + ... + j_{k-l+1} = l$ and $j_1 + 2j_2 + 3j_3 + ... + (k-l+1)j_{k-l+1} = k$. By definition of $\hat{m}_l$ given by (10) and the definition of $m_l$ given in Section 2.3, we have $\hat{m}_l = m_l = 0$ for odd $l$. Consequently, we need only examine terms in the sum such that $j_l = 0$ for all odd $l$, so let us now fix such a term.

For even $l$, we have by Jensen's inequality $m_l = \frac{1}{n}\sum_{i=1}^n \frac{1}{2}E(|\mu_i + \sigma Z|^l) + \frac{1}{2}E(|-\mu_i + \sigma Z|^l) \geq E(|\sigma Z|^l)$ where $Z \sim N(0,1)$. Since $l$ is even, $E(|\sigma Z|^l) = \sigma^l E(Z^l) = \sigma^l \frac{2^{l/2}\Gamma(\frac{l+1}{2})}{\sqrt{\pi}} \geq (c'\sigma^2 l)^{l/2}$ for some small universal constant $c' > 0$ by Stirling's approximation. Consider $(c'\sigma^2 l)^{l/2} \geq \min_{1 \leq q \leq r}(c'\sigma^2 q)^{q/2} \geq C^{-r}(1 \wedge \sigma^r)$ since we take $C > 0$ to be a sufficiently large universal constant.

Observe on the event $\{\max_{1 \leq l \leq r}|\hat{m}_l - m_l| \leq u\}$, we have

$$\left(\frac{\hat{m}_1}{1!}\right)^{j_1}\left(\frac{\hat{m}_2}{2!}\right)^{j_2}\cdots\left(\frac{\hat{m}_{k-l+1}}{(k-l+1)!}\right)^{j_{k-l+1}}$$

$$\leq \left(\frac{m_1 + u}{1!}\right)^{j_1}\left(\frac{m_2 + u}{2!}\right)^{j_2}\cdots\left(\frac{m_{k-l+1} + u}{(k-l+1)!}\right)^{j_{k-l+1}}$$

$$\leq \left(\frac{m_1}{1!}\right)^{j_1}\left(\frac{m_2}{2!}\right)^{j_2}\cdots\left(\frac{m_{k-l+1}}{(k-l+1)!}\right)^{j_{k-l+1}}\exp\left(\sum_{i=1}^{k-l+1} j_i\frac{u}{m_i}\right).$$

To obtain the second line we have used that $0 \leq \left(\frac{\hat{m}_i}{i!}\right)^{j_i} \leq \left(\frac{m_i+u}{i!}\right)^{j_i}$ for all $1 \leq i \leq k-l+1$ since $j_l = 0$ for all odd $l$. We have also used the inequality $1 + x \leq e^x$ to obtain the third line. Since $m_i \geq C^{-r}(1 \wedge \sigma^r)$ for even $i$ as established earlier, it follows by the identity $j_1 + ... + j_{k-l+1} = l$

that $\exp\left(\sum_{i=1}^{k-l+1} j_i \frac{u}{m_i}\right) \le e^{\frac{C^r}{1 \wedge \sigma^r} lu}$. Since $j_1 + 2j_2 + 3j_3 + ... + (k-l+1)j_{k-l+1} = k$ and $j_i = 0$ for all odd $i$, it follows by Jensen's inequality (letting $Y \sim G * N(0, \sigma^2)$) that $m_1^{j_1} \cdots m_{k-l+1}^{j_{k-l+1}} = \prod_{i=1}^{k-l+1} E(Y^i)^{j_i} \le \prod_{i=1}^{k-l+1} E(Y^{ij_i}) = \prod_{i=1}^{k-l+1} E(Y^{k \cdot \frac{ij_i}{k}}) \le \prod_{i=1}^{k-l+1} E(Y^k)^{\frac{ij_i}{k}} = m_k$. This inequality yields

$$\left(\frac{\hat{m}_1}{1!}\right)^{j_1} \left(\frac{\hat{m}_2}{2!}\right)^{j_2} \cdots \left(\frac{\hat{m}_{k-l+1}}{(k-l+1)!}\right)^{j_{k-l+1}}$$
$$\le \left(\frac{m_1}{1!}\right)^{j_1} \left(\frac{m_2}{2!}\right)^{j_2} \cdots \left(\frac{m_{k-l+1}}{(k-l+1)!}\right)^{j_{k-l+1}} + m_k \left(e^{\frac{C^r}{1 \wedge \sigma^r} lu} - 1\right). \tag{16}$$

Let us now prove an analogous lower bound. Again, since $m_i \ge C^{-r}(1 \wedge \sigma^r)$ for even $i$ and $j_1 + ... + j_{k-l+1} = l$, we have

$$\left(\frac{\hat{m}_1}{1!}\right)^{j_1} \left(\frac{\hat{m}_2}{2!}\right)^{j_2} \cdots \left(\frac{\hat{m}_{k-l+1}}{(k-l+1)!}\right)^{j_{k-l+1}}$$
$$= \left(\frac{m_1}{1!}\right)^{j_1} \left(\frac{m_2}{2!}\right)^{j_2} \cdots \left(\frac{m_{k-l+1}}{(k-l+1)!}\right)^{j_{k-l+1}}$$
$$\cdot \left(\left(1 - \frac{m_1 - \hat{m}_1}{m_1}\right)^{j_1} \cdots \left(1 - \frac{m_{k-l+1} - \hat{m}_{k-l+1}}{m_{k-l+1}}\right)^{j_{k-l+1}}\right)$$
$$\ge \left(\frac{m_1}{1!}\right)^{j_1} \left(\frac{m_2}{2!}\right)^{j_2} \cdots \left(\frac{m_{k-l+1}}{(k-l+1)!}\right)^{j_{k-l+1}} \cdot \left(\left(1 - \frac{u}{m_1}\right)^{j_1} \cdots \left(1 - \frac{u}{m_{k-l+1}}\right)^{j_{k-l+1}}\right)$$
$$\ge \left(\frac{m_1}{1!}\right)^{j_1} \left(\frac{m_2}{2!}\right)^{j_2} \cdots \left(\frac{m_{k-l+1}}{(k-l+1)!}\right)^{j_{k-l+1}} \cdot \left(1 - \frac{C^r}{1 \wedge \sigma^r} u\right)^l$$
$$\ge \left(\frac{m_1}{1!}\right)^{j_1} \left(\frac{m_2}{2!}\right)^{j_2} \cdots \left(\frac{m_{k-l+1}}{(k-l+1)!}\right)^{j_{k-l+1}} \cdot \left(1 - \frac{C^r}{1 \wedge \sigma^r} lu\right)$$
$$\ge \left(\frac{m_1}{1!}\right)^{j_1} \left(\frac{m_2}{2!}\right)^{j_2} \cdots \left(\frac{m_{k-l+1}}{(k-l+1)!}\right)^{j_{k-l+1}} - m_k \frac{C^r}{1 \wedge \sigma^r} lu$$
$$\ge \left(\frac{m_1}{1!}\right)^{j_1} \left(\frac{m_2}{2!}\right)^{j_2} \cdots \left(\frac{m_{k-l+1}}{(k-l+1)!}\right)^{j_{k-l+1}} - m_k \left(e^{\frac{C^r}{1 \wedge \sigma^r} lu} - 1\right). \tag{17}$$

Here, we have used $\frac{C^r}{1 \wedge \sigma^r} u \in (0, 1)$, $j_l = 0$ for all odd $l$, and $m_i \ge C^{-r}(1 \wedge \sigma^r)$ for even $i$ to conclude $\left(1 - \frac{m_q - \hat{m}_q}{m_q}\right)^{j_q} \ge \left(1 - \frac{u}{m_q}\right)^{j_q} \ge 0$ for all $q$, thus obtaining the third line. We have also used $\frac{C^r}{1 \wedge \sigma^r} u \in (0, 1)$ to obtain the third-to-last line. Therefore, it follows from (16) and (17) that

$$|\hat{\gamma}_k - \gamma_k| \le \sum_{l=1}^{k} (l-1)! \sum \frac{k!}{j_1! j_2! \cdots j_{k-l+1}!} \cdot m_k \left|e^{\frac{C^r}{1 \wedge \sigma^r} lu} - 1\right|$$
$$\le m_k |e^{\frac{C^r}{1 \wedge \sigma^r} ku} - 1| \cdot (k!)^2 \cdot k^k$$
$$\le 2^k (1 + L^k k!) |e^{\frac{C^r}{1 \wedge \sigma^r} ku} - 1| \cdot (k!)^2 \cdot k^k.$$

The last inequality follows from $||\mu||_\infty \le 1$ and Jensen's inequality. We have also bounded the number of admissible sequences $j_1, ..., j_{k-l+1}$ in the sum by $k^k$ by elementary counting since the constraint $j_i \le k$ must always be satisfied. Maximizing over $1 \le k \le r$ with $k \ne 2$ yields the desired result. $\qquad\square$

Finally, we are able to state a concentration result about $\hat{M}_r$. In the statement of Proposition 7, note the concentration occurs on the same event that the cumulant estimators concentrate around the true cumulants, which in turn occurs, from Proposition 6, on the same event the moment estimators concentrate.

**Proposition 7.** *Suppose $(\mu, \sigma) \in \Theta(L)$ where $\Theta(L)$ is given by (4) and where $L > 0$ is a universal constant. On the event $\left\{ \max_{\substack{1 \leq l \leq r \\ l \neq 2}} |\hat{\gamma}_l - \gamma_l| \leq u \right\}$, we have*

$$\sup_{\gamma \in [0,1]} \left| \hat{M}_r(\gamma) - M_r(\gamma) \right| \leq (2r)^r r^3 (r!)(r^r \vee u^r)u,$$

*where $\hat{M}_r$ and $M_r$ are given by (12) and (8) respectively.*

*Proof.* For $\gamma \in [0,1]$, consider by Definition 1 that

$$\left| \hat{M}_r(\gamma) - M_r(\gamma) \right|$$

$$\leq \sum_{l=1}^{r} \sum \frac{r!}{j_1! j_2! \cdots j_{r-l+1}!} \left| \frac{\gamma}{2!} \right|^{j_2} \left| \prod_{\substack{1 \leq a \leq r-l+1, \\ a \neq 2}} \left( \frac{\hat{\gamma}_a}{a!} \right)^{j_a} - \prod_{\substack{1 \leq a \leq r-l+1, \\ a \neq 2}} \left( \frac{\gamma_a}{a!} \right)^{j_a} \right|$$

$$\leq \sum_{l=1}^{r} \sum \frac{r!}{j_1! j_2! \cdots j_{r-l+1}!} |f_{\mathbf{j}}(\hat{\gamma}) - f_{\mathbf{j}}(\gamma^*)|$$

where the inner sum is taken over all sequences $\mathbf{j} = \{j_1, ..., j_{r-l+1}\}$ of nonnegative integers such that $j_1 + ... + j_{r-l+1} = l$ and $j_1 + 2j_2 + 3j_3 + ... + (r - l + 1)j_{r-l+1} = r$. Here, we have defined the function $f_{\mathbf{j}} : \mathbb{R}^{r-l+1} \to \mathbb{R}$ by $f_{\mathbf{j}}(x) = \prod_{\substack{1 \leq a \leq r-l+1, \\ a \neq 2}} \left( \frac{x_a}{a!} \right)^{j_a}$ and we have used the notation $\hat{\gamma} = (\hat{\gamma}_1, \gamma, \hat{\gamma}_3, ..., \hat{\gamma}_{r-l+1})$ and $\gamma^* = (\gamma_1, \gamma, \gamma_3, ..., \gamma_{r-l+1})$. By Taylor expansion and Holder's inequality, we have

$$|f_{\mathbf{j}}(\hat{\gamma}) - f_{\mathbf{j}}(\gamma^*)| \leq ||\nabla f_{\mathbf{j}}(\xi)||_1 \cdot ||\hat{\gamma} - \gamma^*||_\infty$$

where $\xi$ is some point on the line segment between $\hat{\gamma}$ and $\gamma^*$. It is immediate that on the event $\left\{ \max_{\substack{1 \leq l \leq r \\ l \neq 2}} |\hat{\gamma}_l - \gamma_l| \leq u \right\}$ we have $||\hat{\gamma} - \gamma^*||_\infty \leq u$. Further consider $|\xi_a| \leq |\gamma_a| + u \leq 2(|\gamma_a| \vee u) \leq 2(a^a \vee u)$ by Lemma 9. A straightforward calculation shows

$$||\nabla f_{\mathbf{j}}(\xi)||_1 \leq \sum_{\substack{1 \leq a \leq r-l+1, \\ a \neq 2}} \mathbb{1}_{\{j_a \geq 1\}} j_a \frac{|\xi_a|^{j_a - 1}}{(a!)^{j_a}} \prod_{\substack{1 \leq b \leq r-l+1, \\ b \notin \{2,a\}}} \left| \frac{\xi_b}{b!} \right|^{j_b}$$

$$\leq \sum_{\substack{1 \leq a \leq r-l+1, \\ a \neq 2}} \mathbb{1}_{\{j_a \geq 1\}} j_a \frac{|2(a^a \vee u)|^{j_a - 1}}{(a!)^{j_a}} \prod_{\substack{1 \leq b \leq r-l+1, \\ b \notin \{2,a\}}} \left| \frac{2(b^b \vee u)}{b!} \right|^{j_b}$$

$$\leq 2^r r^2 \left( r^r \vee u^r \right),$$

where we have used $2(a^a \vee u) \geq 1$, $j_a \leq r$, and $\sum_{a=1}^{r-l+1} a j_a = r$. To summarize, we have shown

$$\left| \hat{M}_r(\gamma) - M_r(\gamma) \right| \leq \sum_{l=1}^{r} \sum \frac{r!}{j_1! j_2! \cdots j_{r-l+1}!} \cdot (2^r r^2 (r^r \vee u^r))u \leq (2r)^r r^3 (r!)(r^r \vee u^r)u$$

for all $\gamma \in [0,1]$. As in the proof of Proposition 6, we have bounded the number of admissible sequences $j_1, ..., j_{r-l+1}$ by $r^r$. The proof is complete. $\square$

*Proof of Proposition 4.* Since $(\mu, \sigma) \in \Theta$ and $L > 0$ is a universal constant, it follows by Corollary 1 that the event

$$\mathcal{E} = \left\{ \max_{1 \leq l \leq r} |\hat{m}_l - m_l| \leq \delta^{-1/2} \frac{C_1 e^{C_1' r \log r}}{\sqrt{n}} \right\}$$

has $P_{\mu,\sigma}$-probability of at least $1 - \delta$, where $C_1, C_1' > 0$ are some universal constants. Furthermore, it is clear for any $2 \leq r \leq \frac{1}{C_{\beta,*}} \frac{\log n}{\log \log n}$, we have $\delta^{-1/2} \frac{C_1 e^{C_1' r \log r}}{\sqrt{n}} \lesssim (\log^{\beta/2} n) n^{\frac{C_1'}{C_{\beta,*}} - \frac{1}{2}}$, and so

$C_{\beta,*}$ can be taken sufficiently large to ensure $\delta^{-1/2}\frac{C_1 e^{C_1' r \log r}}{\sqrt{n}}$ decays at some rate that is polynomial in $n$. We will now apply Proposition 6 on the event $\mathcal{E}$ with $u = \delta^{-1/2}\frac{C_1 e^{C_1' r \log r}}{\sqrt{n}}$. To use Proposition 6, it must be checked $\frac{\tilde{C}^r}{1 \wedge \sigma^r} u \in (0,1)$ where $\tilde{C} > 0$ is the universal constant from Proposition 6. Consider that $\sigma^2 \geq \frac{c^*}{r}$, and so

$$
\frac{\tilde{C}^r}{1 \wedge \sigma^r} u \leq \left( \tilde{C}^r \vee \left( \frac{\tilde{C}}{\sqrt{c^*}} \right)^r r^{r/2} \right) u
$$

$$
\leq \delta^{-1/2} \frac{C_1}{\sqrt{n}} \exp\left( \left( C_1' + \frac{1}{2} \right) r \left( \log r + \log\left( \frac{\tilde{C}}{\sqrt{c^*}} \right) + \log \tilde{C} \right) \right)
$$

$$
\lesssim (\log^{\beta/2} n) n^{\frac{C_1' + 1/2 + \log(\tilde{C}^2/\sqrt{c^*})}{C_{\beta,*}} - \frac{1}{2}}.
$$

Therefore, taking $C_{\beta,*}$ sufficiently large (depending on $c^*$ and $\beta$) implies $\frac{\tilde{C}^r}{1 \wedge \sigma^r} u$ decays at some rate that is polynomial in $n$. It then follows from Proposition 6, the fact that $r$ is at most logarithmic in $n$, and the inequality $|e^x - 1| \leq ex$ for $x \in (0,1)$,

$$
\max_{\substack{1 \leq l \leq r, \\ l \neq 2}} |\hat{\gamma}_l - \gamma_l| \leq (1 + L^r r!) \cdot (r!)^2 \cdot (2r)^r \cdot \left| e^{\frac{\tilde{C}^r}{1 \wedge \sigma^r} r u} - 1 \right| \leq C_2 e^{C_2' r \log r} u
$$

where $C_2, C_2' > 0$ are some universal constants (potentially depending on $c^*$). Note $C_2 e^{C_2' r \log r} u \lesssim (\log^{\beta/2} n) n^{\frac{C_1' + C_2'}{C_{\beta,*}} - \frac{1}{2}}$, and so again we can take $C_{\beta,*}$ sufficiently large to ensure $C_2 e^{C_2' r \log r} u$ decays at some rate that is polynomial in $n$. It then follows from Proposition 7 that

$$
\sup_{\gamma \in [0,1]} \left| \hat{M}_r(\gamma) - M_r(\gamma) \right| \leq C_3 e^{C_3' r \log r} \cdot \left( C_2 e^{C_2' r \log r} u \right) = \delta^{-1/2} \frac{C_1 C_2 C_3 e^{(C_1' + C_2' + C_3') r \log r}}{\sqrt{n}},
$$

where $C_3, C_3' > 0$ are some universal constants (potentially depending on $c^*$). The proof is complete.

$\square$

## B.3 Proof of Theorem 1

*Proof of Theorem 1.* Fix $(\mu, \sigma) \in \Theta$. Choose

$$
C^* := C_{\beta,*} \vee 2 \left( 4(C'+1) \right) \vee 2 \cdot \inf\left\{ K > 0 : \left( \frac{4(C'+1)}{K} - 1 \right) \cdot 2K < -2 \right\}.
$$

Note $C^*$ is a positive universal constant since $C_{\beta,*}$, $C$, and $C'$ all are positive universal constants. The reason for this choice of $C^*$ will become clear later on. To bound the estimation risk, first consider

$$
E_{\mu,\sigma} \left( |\hat{\sigma}^2 - \sigma^2|^2 \right) \leq \sigma^4 P_{\mu,\sigma} \left\{ \max_{1 \leq i \leq n} X_i \leq 4\sqrt{\frac{\log n}{r}} \right\} \tag{18}
$$

$$
+ E_{\mu,\sigma} \left( \left| \frac{1}{n} \sum_{i=1}^n X_i^2 - \hat{\gamma}_2(r) - \sigma^2 \right|^2 \mathbb{1}_{\left\{ \max_{1 \leq i \leq n} X_i > 4\sqrt{\frac{\log n}{r}} \right\}} \right). \tag{19}
$$

To bound (18), let $Z_i = X_i - \mu_i$ and consider by Lemma 10,

$$
\sigma^4 P_{\mu,\sigma} \left\{ \max_{1 \leq i \leq n} X_i \leq 4\sqrt{\frac{\log n}{r}} \right\}
$$

$$
\leq \sigma^4 P_{\mu,\sigma} \left\{ \left| \max_{1 \leq i \leq n} Z_i - E_{\mu,\sigma} \left( \max_{1 \leq i \leq n} Z_i \right) \right| \geq E_{\mu,\sigma} \left( \max_{1 \leq i \leq n} Z_i \right) - 4\sqrt{\frac{\log n}{r}} - ||\mu||_\infty \right\}
$$

$$
\leq 2\sigma^4 \exp \left( -\frac{\left( E_{\mu,\sigma} (\max_{1 \leq i \leq n} Z_i) - 4\sqrt{\frac{\log n}{r}} - ||\mu||_\infty \right)_+^2}{2\sigma^2} \right)
$$

$$
\leq 2\sigma^4 \exp \left( -\frac{1}{2} \left( \frac{\sqrt{\log n}}{\sqrt{\pi \log 2}} - \sigma^{-1} \left( 4\sqrt{\frac{\log n}{r}} + 1 \right) \right)_+^2 \right)
$$

$$
\lesssim \left( \frac{\log \log n}{\log n} \right)^2 \tag{20}
$$

where we have used the result of [32] which gives $E_{\mu,\sigma}(\max_{1 \leq i \leq n} Z_i) \geq \frac{\sigma}{\sqrt{\pi \log 2}} \sqrt{\log n}$. We have also used $||\mu||_\infty \leq 1$ and $\frac{1}{r} \lesssim \frac{\log \log n}{\log n}$. To obtain the result of the theorem, it remains to show (19) has order at most $\left( \frac{\log \log n}{\log n} \right)^2$. To do so, we split the analysis into two cases.

**Case 1:** Suppose $\sigma^2 < \frac{1}{r}$. Consider by Cauchy-Schwarz inequality,

$$
(19) \leq \sqrt{E_{\mu,\sigma} \left( \left| \frac{1}{n} \sum_{i=1}^n X_i^2 - \hat{\gamma}_2(r) - \sigma^2 \right|^4 \right) \cdot P_{\mu,\sigma} \left\{ \max_{1 \leq i \leq n} X_i > 4\sqrt{\frac{\log n}{r}} \right\}}. \tag{21}
$$

Letting $Z_i = X_i - \mu_i$, consider by Lemma 10 we have

$$
P_{\mu,\sigma} \left\{ \max_{1 \leq i \leq n} X_i > 4\sqrt{\frac{\log n}{r}} \right\}
$$

$$
\leq P_{\mu,\sigma} \left\{ \left| \max_{1 \leq i \leq n} Z_i - E_{\mu,\sigma} \left( \max_{1 \leq i \leq n} Z_i \right) \right| > 4\sqrt{\frac{\log n}{r}} - ||\mu||_\infty - E_{\mu,\sigma} \left( \max_{1 \leq i \leq n} Z_i \right) \right\}
$$

$$
\leq 2 \exp \left( -\frac{1}{2\sigma^2} \left( 4\sqrt{\frac{\log n}{r}} - ||\mu||_\infty - E_{\mu,\sigma} \left( \max_{1 \leq i \leq n} Z_i \right) \right)_+^2 \right).
$$

Consider $E_{\mu,\sigma} (\max_{1 \leq i \leq n} Z_i) \leq \sigma\sqrt{2 \log n} \leq \sqrt{\frac{2 \log n}{r}}$ and $||\mu||_\infty \leq 1$. Therefore, it follows that $P_{\mu,\sigma} \left\{ \max_{1 \leq i \leq n} X_i > 4\sqrt{\frac{\log n}{r}} \right\} \leq 2 \exp \left( -\frac{\log n}{r\sigma^2} \right) \leq \frac{2}{n}$. With this in hand, it follows from $E_{\mu,\sigma} \left( \left| \frac{1}{n} \sum_{i=1}^n X_i^2 - \hat{\gamma}_2(r) - \sigma^2 \right|^4 \right) \lesssim 1$ that (21) $\lesssim n^{-1} \lesssim \left( \frac{\log \log n}{\log n} \right)^2$ as desired.

**Case 2:** Suppose $\sigma^2 \geq \frac{1}{r}$. It is clear (19) $\leq E_{\mu,\sigma} \left( \left| \frac{1}{n} \sum_{i=1}^n X_i^2 - \hat{\gamma}_2(r) - \sigma^2 \right|^2 \right)$. To bound this expectation, denote the event $\mathcal{E} = \left\{ \sup_{\gamma \in [0,1]} \left| \hat{M}_r(\gamma) - M_r(\gamma) \right| \leq \varepsilon \right\}$. Since $\sigma^2 \geq \frac{1}{r}$, $C^* \geq C_{\beta,*}$, and $\varepsilon = \frac{Ce^{C'r\log r}}{\sqrt{n\delta}}$ with $C, C'$ as in Proposition 4 with the choice $c^* = 1$, it follows by Proposition 4 that $P_{\mu,\sigma}(\mathcal{E}) \geq 1 - \delta$. From the inequality $(a + b)^2 \leq 2a^2 + 2b^2$, it follows $\left| \frac{1}{n} \sum_{i=1}^n X_i^2 - \hat{\gamma}_2(r) - \sigma^2 \right|^2 \leq 2 \left| \frac{1}{n} \sum_{i=1}^n X_i^2 - (\gamma_2 + \sigma^2) \right|^2 + 2|\hat{\gamma}_2(r) - \gamma_2|^2$. Since $\hat{\gamma}_2(r), \gamma_2 \in$

$[0, 1]$ and since $\varepsilon = \frac{Ce^{C'r\log r}}{\sqrt{n\delta}}$, an application of Proposition 3 yields

$$2|\hat{\gamma}_2(r) - \gamma_2|^2 \leq 2\left(\frac{2Ce^{(C'+1)r\log r}}{\sqrt{n\delta}}\right)^{4/r} + \frac{C''}{r^2} + 2\mathbb{1}_{\mathcal{E}^c}$$

where $C'' > 0$ is a universal constant. Consider

$$\left(\frac{e^{(C'+1)r\log r}}{\sqrt{n\delta}}\right)^{4/r} = \exp\left(\frac{1}{r}\left(4(C'+1)r\log r - 2\log n + 4\log\log n\right)\right)$$

$$\leq \exp\left(\frac{1}{r}\left(\frac{4(C'+1)}{C^*}\frac{\log n}{\log\log n}\log\left(\frac{1}{C^*}\frac{\log n}{\log\log n}\right) - \log n\right)\right)$$

$$\leq \exp\left(\left(\frac{4(C'+1)}{C^*} - 1\right)\frac{\log n}{r}\right).$$

Note $C^* \geq 2 \cdot 4(C'+1) \geq 4(C'+1)$, so it immediately follows $\frac{4(C'+1)}{C^*} - 1 < 0$. Therefore, it follows from $r \geq \frac{1}{2C^*}\frac{\log n}{\log\log n}$ that

$$\left(\frac{e^{(C'+1)r\log r}}{\sqrt{n\delta}}\right)^{4/r} \leq \exp\left(\left(\frac{4(C'+1)}{C^*} - 1\right) \cdot 2C^*\log\log n\right) \leq \frac{1}{\log^2 n}$$

where the second inequality follows from the fact that $C^*$ satisfies, by design, the inequality $\left(\frac{4(C'+1)}{C^*} - 1\right) \cdot 2C^* \leq -2$. Therefore, we have shown $2\left(\frac{2Ce^{(C'+1)r\log r}}{\sqrt{n\delta}}\right)^{4/r} \lesssim \frac{1}{\log^2 n}$. To summarize, we have shown

$$E_{\mu,\sigma}\left(\left|\frac{1}{n}\sum_{i=1}^n X_i^2 - \hat{\gamma}_2(r) - \sigma^2\right|^2\right) \lesssim \mathrm{Var}\left(\frac{1}{n}\sum_{i=1}^n X_i^2\right) + E_{\mu,\sigma}\left(|\hat{\gamma}_2(r) - \gamma_2|^2\right)$$

$$\lesssim \frac{1}{n} + \frac{1}{\log^2 n} + \frac{1}{r^2} + P_{\mu,\sigma}(\mathcal{E}^c)$$

$$\lesssim \frac{1}{n} + \frac{1}{\log^2 n} + \frac{1}{r^2} + \delta$$

$$\asymp \left(\frac{\log\log n}{\log n}\right)^2$$

as desired. The proof is complete. $\qquad\square$

## C   Lower bound

*Proof of Theorem 2.* As seen in many minimax lower bound arguments of the literature, we proceed by reducing to a two-point testing problem. Without loss of generality we will assume $n$ is larger than a sufficiently large universal constant. Let $0 < \tau^2 \leq 1$ and we will choose it later. Set $\sigma_0^2 = 1 + \tau^2$ and $\sigma_1^2 = 1$. For a distribution $\pi$ supported on $[-1, 1]$, denote the induced mixture $P_{\pi,\sigma} = \int P_{\mu,\sigma}\,\pi^{\otimes n}(d\mu)$. It follows by reverse triangle inequality

$$\inf_{\hat{\sigma}}\sup_{(\mu,\sigma)\in\Theta(2)} P_{\mu,\sigma}\left\{|\hat{\sigma}^2 - \sigma^2| \geq \frac{\tau^2}{2}\right\}$$

$$\geq \frac{1}{2}\inf_{\hat{\sigma}}\left\{P_{0,\sigma_0}\left\{|\hat{\sigma}^2 - \sigma_0^2| \geq \frac{\tau^2}{2}\right\} + P_{\pi,\sigma_1}\left\{|\hat{\sigma}^2 - \sigma_1^2| \geq \frac{\tau^2}{2}\right\}\right\}$$

$$\geq \frac{1}{2}\inf_{\hat{\sigma}}\left\{P_{0,\sigma_0}\left\{|\hat{\sigma}^2 - 1| \leq \frac{\tau^2}{2}\right\} + P_{\pi,\sigma_1}\left\{|\hat{\sigma}^2 - 1| > \frac{\tau^2}{2}\right\}\right\}$$

$$\geq \frac{1}{2}\inf_{\mathcal{A}}\left\{P_{0,\sigma_0}(\mathcal{A}) + P_{\pi,\sigma_1}(\mathcal{A}^c)\right\}$$

$$= \frac{1}{2}\left(1 - \mathrm{d}_{\mathrm{TV}}(P_{0,\sigma_0}, P_{\pi,\sigma_1})\right), \tag{22}$$

where the infimum in the penultimate line runs over all events $\mathcal{A}$. By the Neyman-Pearson lemma, $1 - \mathrm{d}_{\mathrm{TV}}(P_{0,\sigma_0}, P_{\pi,\sigma_1})$ is the optimal testing risk for the hypothesis testing problem

$$H_0 : X_1, ..., X_n \overset{iid}{\sim} N(0, 1 + \tau^2),$$

$$H_1 : X_1, ..., X_n \overset{iid}{\sim} \pi * N(0, 1).$$

Specifically, we have $\mathrm{d}_{\mathrm{TV}}(P_{0,\sigma_0}, P_{\pi,\sigma_1}) \leq \frac{1}{2}\sqrt{(1 + \chi^2(\pi * N(0,1) \,\|\, N(0, 1 + \tau^2)))^n - 1}$ by Lemma 5. Hence, it suffices to bound the $\chi^2$-divergence to furnish a lower bound for (22).

We now construct $\pi$ and we will pick $\tau^2 < \frac{1}{16}$ at the end of the proof to obtain the claimed lower bound. Let $D$ denote the largest even number smaller than or equal to $\frac{1}{4\tau^2}$. Note $D \geq 2$ since $\tau^2 < \frac{1}{8}$. Let $g_D = \sum_{i=1}^{D} w_i \delta_{z_i}$ denote the $D$-point Gaussian quadrature of $N(0, 1)$. Note that the atoms $\{z_i\}_{i=1}^{D}$ are the zeros of the $D$th degree Hermite polynomial (Lemma 6). Further note that this measure is 1-subgaussian and symmetric about zero as $D$ is even (Lemma 7). Take the prior distribution $\pi = \sum_{i=1}^{D} w_i \delta_{\tau z_i}$. In order for $\pi$ to be a valid choice, it must be verified $\pi$ is supported on $[-1, 1]$. Lemma 8 gives us that $|z_i| \leq \sqrt{4D - 4}$ since $z_i$ is a zero of the $D$th degree Hermite polynomial. Observe $|\tau z_i| \leq \sqrt{\tau^2(4D - 4)} < 1$ by our choice of $D$. Hence, $\pi$ is supported on $[-1, 1]$ and thus is a valid choice.

We use a moment matching technique (Proposition 8). To do so, we first verify the first $2D - 1$ moments of $\pi$ and $N(0, \tau^2)$ match. For any $r \in \{1, ..., 2D - 1\}$, observe since $g_D$ is the $D$-point Gaussian quadrature of $N(0, 1)$, it follows $E_{Y \sim N(0,\tau^2)}(Y^r) = \tau^r E_{Z \sim N(0,1)}(Z^r) = \tau^r E_{Z \sim g_D}(Z^r) = E_{Y \sim \pi}(Y^r)$. Note both $N(0, \tau^2)$ and $\pi$ are $\tau$-subGaussian. Proposition 8, along with $D \geq \frac{1}{4\tau^2} - 2 \geq 2$ and $\tau^2 < \frac{1}{16}$, thus implies

$$\chi^2(\pi * N(0,1) \,\|\, N(0, \tau^2) * N(0,1)) \leq \frac{16\tau^{4D}}{\sqrt{2D - 1}(1 - \tau^2)} \leq \frac{16^2}{15\sqrt{3}} \exp\left(-\frac{1}{4\tau^2} \log\left(\frac{1}{\tau^2}\right)\right).$$

Select $\tau^2 = \frac{\log\log n}{16 \log n}$ and observe

$$\chi^2(\pi * N(0,1) \,\|\, N(0, \tau^2) * N(0,1)) \leq \frac{16^2}{15\sqrt{3}} \exp\left(-4\log n\left(1 - \frac{\log\left(16^{-1}\log\log n\right)}{\log\log n}\right)\right) \leq \frac{1}{n}$$

since $n$ is larger than a sufficiently large universal constant. Therefore, $\mathrm{d}_{\mathrm{TV}}(P_{0,\sigma_0}, P_{\pi,\sigma_1}) \leq \frac{1}{2}\sqrt{\left(1 + \frac{1}{n}\right)^n - 1} \leq \frac{1}{2}\sqrt{e - 1}$, which, when plugged into (22), yields the lower bound

$$\inf_{\hat{\sigma}} \sup_{(\mu,\sigma) \in \Theta(2)} P_{\mu,\sigma}\left\{|\hat{\sigma}^2 - \sigma^2| \geq \frac{\log\log n}{32 \log n}\right\} \geq \frac{1}{2}\left(1 - \frac{1}{2}\sqrt{e - 1}\right).$$

The proof is complete. $\qquad\qquad\qquad\qquad\qquad\qquad\qquad\qquad\qquad\qquad\qquad\qquad\square$

## D    Auxiliary definitions and results

**Definition 2.** *A probability distribution $P$ is said to be $a$-subgaussian for $a > 0$ if for $X \sim P$ we have $E\left(\exp\left(\lambda(X - E(X))\right)\right) \leq e^{\frac{\lambda^2 a^2}{2}}$ for all $\lambda \in \mathbb{R}$.*

**Lemma 5** ($\chi^2$ tensorization [58]). *If $P = \bigotimes_{i=1}^{n} P_i$ and $Q = \bigotimes_{i=1}^{n} Q_i$ are product measures, then $\chi^2(P \,\|\, Q) = \prod_{i=1}^{n}(1 + \chi^2(P_i \,\|\, Q_i)) - 1$.*

**Proposition 8** (Theorem 3.3.3 [64]). *Suppose $\nu$ and $\nu'$ are two symmetric probability distributions that are $\varepsilon$-subgaussian for $\varepsilon < 1$. If the first $D$ moments of $\nu$ and $\nu'$ are equal, then $\chi^2(\nu * N(0,1) \,\|\, \nu' * N(0,1)) \leq \frac{16}{\sqrt{D}} \frac{\varepsilon^{2D+2}}{1-\varepsilon^2}$.*

**Proposition 9** (Gaussian quadrature [64]). *Suppose $\mu$ is a probability measure supported on $E \subset \mathbb{R}$. If $k \geq 1$, there exists a $k$-atomic distribution $\mu_k = \sum_{i=1}^{k} w_i \delta_{x_i}$ supported on $E$ such that for any polynomial $p$ of degree at most $2k - 1$ we have $\int p(x) \, d\mu(x) = \sum_{i=1}^{k} w_i p(x_i)$.*

**Lemma 6** (Remark 2.7.2 [64]). *If $k \geq 1$, then the $k$-point Gaussian quadrature of $N(0,1)$ has its atoms at the roots of the degree $k$ Hermite polynomial $He_k$.*

**Lemma 7** (Lemma 2.7.3 [64]). *Suppose $k \geq 1$ and $g_k$ is the $k$-point Gaussian quadrature of $N(0,1)$. For $j \geq 2k$, we have $E_{X \sim g_k}(X^j) \leq E_{Z \sim N(0,1)}(Z^j)$ when $j$ is even and $E_{X \sim g_k}(X^j) = 0$ otherwise. In particular, $g_k$ is 1-subgaussian.*

**Lemma 8.** *If $k \geq 1$, then the zeros of the $k$th Hermite polynomial lie in $[-\sqrt{4k-4}, \sqrt{4k-4}]$.*

*Proof.* From (6.2.18) on page 120 of [56] the zeros of the $k$th degree physicist's Hermite polynomial $H_k(x) = (-1)^n e^{x^2} \frac{d^k}{dx^k} e^{-x^2}$ lie in the interval $\left[ -\frac{\sqrt{2}(k-1)}{\sqrt{k+2}}, \frac{\sqrt{2}(k-1)}{\sqrt{k+2}} \right]$. Since $k - 1 \leq k + 2$, it immediately follows that we have the inclusion $\left[ -\frac{\sqrt{2}(k-1)}{\sqrt{k+2}}, \frac{\sqrt{2}(k-1)}{\sqrt{k+2}} \right] \subset \left[ -\sqrt{2k-2}, \sqrt{2k-2} \right]$. Since we have the following correspondence between physicist's and probabilist's Hermite polynomials $H_k(x) = 2^{k/2} He_k(\sqrt{2}x)$, it follows the zeros of $He_k$ lie in $[-\sqrt{4k-4}, \sqrt{4k-4}]$. The proof is complete. $\qquad\square$

**Lemma 9.** *The $r$th cumulant $\gamma_r$ of a distribution $G$ supported on $[-1,1]$ satisfies $|\gamma_r| \leq r^r$.*

*Proof.* Let $\nu_k$ denote the $k$th moment of $G$ and note $|\nu_k| \leq 1$. From (6) it follows $\gamma_r = \sum_{l=1}^{r} (-1)^{l-1}(l-1)! B_{r,l}(\nu_1, \nu_2, ..., \nu_{r-l+1})$. Since the coefficients of the Bell polynomial $B_{r,l}$ are all positive and $|\nu_k| \leq 1$, it follows $|\gamma_r| \leq \sum_{l=1}^{r}(l-1)! B_{r,l}(1,1,...,1) = \sum_{l=1}^{r}(l-1)! \left\{ {r \atop l} \right\}$ where $\left\{ {r \atop l} \right\}$ is a Stirling number of the second kind. Consider $(l-1)! \left\{ {r \atop l} \right\} = \frac{1}{l} \cdot l! \left\{ {r \atop l} \right\} \leq l^{r-1}$. Therefore, $|\gamma_r| \leq \sum_{l=1}^{r} l^{r-1} \leq r^r$ as desired. $\qquad\square$

**Lemma 10** (Lemma 2.10.6 [57]). *If $Z \sim N(0, \sigma^2 I_n)$, then*

$$P \left\{ \left| \max_{1 \leq i \leq n} Z_i - E \left( \max_{1 \leq i \leq n} Z_i \right) \right| \geq u \right\} \leq 2 \exp \left( -\frac{u^2}{2\sigma^2} \right)$$

*for $u \geq 0$.*

## E  Notation

For $a, b \in \mathbb{R}$ the notation $a \lesssim b$ denotes the existence of a universal constant $c > 0$ such that $a \leq cb$. The notation $a \gtrsim b$ is used to denote $b \lesssim a$. Additionally $a \asymp b$ denotes $a \lesssim b$ and $a \gtrsim b$. The symbol $:=$ is frequently used when defining a quantity or object. Furthermore, we frequently use $a \vee b := \max(a, b)$ and $a \wedge b := \min(a, b)$. We generically use the notation $\mathbb{1}_A$ to denote the indicator function for an event $A$. For two probability measures $P$ and $Q$ on a measurable space $(\mathcal{X}, \mathcal{A})$, the total variation distance is defined as $d_{\mathrm{TV}}(P, Q) := \sup_{A \in \mathcal{A}} |P(A) - Q(A)|$. If $P$ is absolutely continuous with respect to $Q$, then the $\chi^2$-divergence is defined as $\chi^2(P||Q) := \int_{\mathcal{X}} \left( \frac{dP}{dQ} - 1 \right)^2 dQ$. We will frequently use the same notation for two probability densities $p$ and $q$. For sequences $\{a_k\}_{k=1}^{\infty}$ and $\{b_k\}_{k=1}^{\infty}$, the notation $a_k = o(b_k)$ denotes $\lim_{k \to \infty} \frac{a_k}{b_k} = 0$ and the notation $a_k = \omega(b_k)$ is used to denote $b_k = o(a_k)$. For a point $x \in \mathbb{R}$, the symbol $\delta_x$ denotes the probability measure which places full probability mass at the point $x$. The symbol $*$ denotes convolution and the same symbol will be used in the context of the convolution of probability measures as well as functions. Throughout, iterated logarithms will be used (e.g. expressions like $\log \log n$). Without explicitly stating so, we will take such an expression to be equal to some universal constant if otherwise it would be less than one. For example, $\log \log n$ should be understood to be equal to a universal constant when $n < e^e$.

