# OpenReview forum: "Variance estimation in compound decision theory under boundedness"
_NeurIPS.cc/2024/Conference — NeurIPS 2024 poster_

### Official Review · Reviewer_RbL1 · 2024-07-09

**Soundness:** 4
**Presentation:** 4
**Contribution:** 3
**Rating:** 7
**Confidence:** 4

**Summary:**

This submission studies the variance estimation problem for Gaussians under the compound decision/empirical Bayes settings, assuming bounded means and variance. The main results are up-to-constants matching upper and lower bounds on the estimation rate, for mean squared error. For the upper bound, the proposed algorithm is based on estimating cumulants and using Bell polynomials to find the cumulant $\gamma_2$, which is almost directly related to the desired variance parameter. For the lower bound, the authors use a moment matching and $\chi^2$-divergence approach to show indistinguishability results.

**Strengths:**

The paper is very very well-written, with intuition properly explained and motivated. It is refreshing to see a properly-written paper being submitted to NeurIPS/ICML.

Technically, it is also nice that the authors settle the estimation rate up to constants for this problem. I also find the cumulant-based approach interesting --- I don't think I've seen many papers (at least in recent memory) that forego moments and deal with cumulants instead.

**Weaknesses:**

Here are some constructive criticisms that I hope the authors will find useful to improve the paper.

- The title: I think it is a little too general and perhaps slightly misleading, given that there has been a lot of other works studying the Gaussian variance estimation problem in the compound decisions setting, under other assumptions. The authors should consider including something to the effect of "under bounded means/variance assumption".

- In Section 3, consider re-ordering the explanation. Lines 340-341 says "This can be achieved by constructing $G$ to share a large number of moments with $\mathcal{N}(0,\tau^2)$." without much explanation. For readers who haven't seen this kind of approach, it would be helpful to explain immediately to them why this moment matching is relevant. The prose on $\chi^2$-divergence etc. later on should be moved up in my opinion, instead of diving straight into a discussion on how many moments one can match.

- My main criticism is actually that I don't find the problem setting super well-motivated, although I don't necessarily hold this against this submission in particular. I understand that this problem is very well-studied in the literature under different assumptions. However, for one, I wish the motivation were more explicit in this submission, since I'm not familiar with the model until reading this paper. On top of that, perhaps more importantly, when I first saw problem definition on the first page, my immediate thought was "ah, this is *really* going to depend on the Gaussian noise isn't it", and had pretty much the same identifiability doubts as what turned out to be discussed in Section 4. Given that information-theoretically one pretty much needs to have quite a bit of information about the noise distribution, I'm not too convinced about the applicability of the results. Having said that, I again understand that the problem has been well-studied in the literature, so clearly there are others who care about the problem, so I won't hold this against this paper in particular (as evidenced by my "Accept" score).

**Questions:**

- The parameter space $\Theta(L)$, I presume the estimator (through the many hidden constants in propositions and theorems) depend on the knowledge of what $L$ is? My next question is, how does the estimation rate depend on $L$? Similarly, how does the estimation rate depend on the bound on the $\ell_\infty$ norm of $\mu$?

- What is the concentration behavior/tails of the estimator? Theorem 1 is in mean squared error only.

- Line 283 mentions that $\hat{M}_r$ concentrates around the expectation $M_r$ uniformly over the input $\gamma$. Can you comment roughly on the technique to show that, or at least why it deserves to be true? I think it's useful to comment on that even in the main paper, since uniform concentration is usually an annoying thing to show, if not sometimes a technical gist of a problem.

**Limitations:**

Yes.

---

> ### Author Rebuttal · Authors · 2024-08-03
>
> Thank you for your very helpful comments and thorough feedback! Our responses below address the points raised in your review.
>
> [Weakness 1] We agree the title can be revised for clarity. In the revision, we will revise the title to ``Variance estimation in compound decision theory under boundedness''.
>
> [Weakness 2] We agree the explanation would improve after the suggested re-ordering. We will do this in the revision.
>
> [Weakness 3] This comment is acknowledged and well-taken. Indeed, we are leaning heavily on statistical tradition and the existence of a substantial literature to motivate consideration of the problem studied in our article, and especially so to justify the Gaussian assumption.
>
> [Question 1] Yes, our estimator relies on knowledge of $L$. Please see the global response to all reviewers regarding the dependence of the rate on $L$. The question regarding the rate's dependence on the bound on the $\ell_\infty$ norm of $\mu$ is very related. Suppose we are in the setting where we know $||\mu||\_\infty \leq B$ and $\sigma \leq L$. We can transform the data without loss of any information as $Y_i = X_i/B$, and so we have $Y_i \overset{ind}{\sim} N(\theta_i, \tau^2)$ where $\theta_i = \mu_i/B$ and $\tau = \sigma/B$. Now note $||\theta||_\infty \leq 1$ and $\tau \leq L/B$, and so we are in the original setting of our paper. Estimation of $\tau^2$ from $Y$ is equivalent to estimation of $\sigma^2$ from $X$ (up to an appropriate scaling). For the same reasons outlined in the global response, estimation of $\tau^2$ is essentially only interesting in the case $L/B \asymp 1$. In this case, the minimax rate of estimation of $\tau^2$ is $(\log\log n / \log n)^2$, which thus implies the minimax rate of estimating $\sigma^2$ is $B^4 (\log\log n/\log n)^2$.
>
> [Question 2] The question regarding the concentration behavior of the estimator appears to us to be quite tricky. As can be surmised from Proposition 3, the fluctuations of the error $|\hat{\gamma}\_2(r) - \gamma\_2|$ can be related to the fluctuations of $\sup\_{\gamma \in [0, 1]} |\hat{M}\_r(\gamma) - M\_r(\gamma)|$. The latter random variable involves polynomials of growing degree (since we choose $r$ to grow), and so it appears difficult to us to obtain concentration results beyond that obtained by a simplistic analysis using very crude tools.
>
> [Question 3] It turns out the structure of the Bell polynomials makes showing uniform concentration very painless. The relevant analysis occurs in the proof of Proposition 7. In particular, recalling definitions from our paper, consider $|\hat{M}\_r(\gamma) - M\_r(\gamma)| \leq \sum_{l=1}^{r} |B\_{r, l}(\hat{\gamma}\_1, \gamma, \hat{\gamma}\_3,...,\hat{\gamma}\_{r-l+1}) - B_{r, l}(\gamma\_1, \gamma, \gamma\_3,...,\gamma\_{r-l+1})| \leq \sum_{l=1}^{r} \sum \frac{r!}{j\_1! j\_2! \cdot\cdot\cdot j\_{r-l+1}!} \left|\frac{\gamma}{2!}\right|^{j\_2} \cdot W\_{\mathbf{j}}$ where $W\_{\mathbf{j}}$ is a random variable which does not depend on $\gamma$. Here, the inner sum is taken over appropriate sequences $\mathbf{j} = \{j\_1,j\_2,...,j\_{r-l+1}\}$ as in the definition of Bell polynomials (Definition 1). Therefore, it is immediate to obtain the deterministic inequality $\sup\_{\gamma \in [0, 1]} |\hat{M}\_r(\gamma) - M\_r(\gamma)| \leq \sum\_{l=1}^{r} \sum \frac{r!}{j\_1! j\_2! \cdot\cdot\cdot j\_{r-l+1}!}\cdot W\_{\mathbf{j}}$, and now the right hand side no longer involves a supremum. The Bell polynomial structure is very convenient. In the revision, we will make a comment about it in the main text.

---

> > ### Comment · Reviewer_RbL1 · 2024-08-07
> >
> > Thanks for your responses. I'll maintain my current score, and I hope the authors will incorporate the promised revisions, including the explanation of the scaling with $L$ and $B$ at least in the appendix (fleshed out, even if it's a straightforward rescaling argument).
> >
> > By the way, for the dependence on $B$, do you mean $B^2$ instead of $B^4$ or am I misreading/misunderstanding? It's just a linear rescaling of the variable with a factor of $B$, so it should change the variance by $B^2$?

---

> > > ### Author Response · Authors · 2024-08-10
> > >
> > > Thank you very much for reviewing our responses. As you correctly note, it is a linear rescaling with a factor of $B$, which scales the variance by $B^2$. However, when speaking of the "rate", it is referring to squared error, so squaring yields a factor of $B^4$.

---

> > > > ### Comment · Reviewer_RbL1 · 2024-08-10
> > > >
> > > > Thank you; I forgot that this submission phrases things in the squared error for the variance.

---

### Official Review · Reviewer_9ofA · 2024-07-09

**Soundness:** 4
**Presentation:** 3
**Contribution:** 4
**Rating:** 7
**Confidence:** 3

**Summary:**

This paper gives a sharp minimax rate of the variance estimation under mild assumption on a Gaussian-based model.

**Strengths:**

The theoretical results are solid and complete, and the related reference are discussed and linked with their own work.

**Weaknesses:**

The sign in equation (5) and also in line 94 in not clear for me, does it means proportional to? This paper is pretty theortical-based one, and it would be better if there are some experiments.

**Questions:**

N/A

---

> ### Author Rebuttal · Authors · 2024-08-03
>
> Thank you for your feedback! Our response to your review is below.
>
> [Weakness 1] We have conducted an experiment in the attached pdf, and the details are in the global response to all reviewers. Yes, the symbols $\asymp$ and $\lesssim$ mean ``up to universal constants''. The notation we use is defined in Appendix E, and in the revision we will note this in the main text to make the reader aware.

---

> > ### Comment · Reviewer_9ofA · 2024-08-09
> > **Response to rebuttal**
> >
> > I thank the authors for their rebuttal. It is good to see the experiment is done, but a larger/complex problem with real-world data is more preferable. Thanks for taking my suggestions on the symbols and moving the definition to the main text. Overall, I will keep my score.

---

### Official Review · Reviewer_tWCj · 2024-07-13

**Soundness:** 3
**Presentation:** 3
**Contribution:** 2
**Rating:** 5
**Confidence:** 2

**Summary:**

This paper studies the problem of variance estimation in compound decision setting, with the assumption that means are bounded. Main contributions:

- The authors prove the minimax rate of variance estimation in the setting of the paper, with the proof of the lower bound and a proposed estimator achieving the convergence rate.
- The authors discuss the importance of the Gaussian assumption of the result.

Post rebuttal: I acknowledge that I have read the authors responses and other reviewers' comments.

**Strengths:**

Strengths:
- This paper is well-written and well-organized.
- The authors provide the minimax rate of variance estimation in the setting that means are bounded and noise follow the Gaussian distribution.
- The authors discuss the noise agnosticism that the estimator is designed for Gaussian noises.

**Weaknesses:**

Weaknesses:
- There is no discussions on the computation complexity of the proposed estimator, and no experiments.

**Questions:**

Questions:
- In real applications (where $\mu's$ are not chosen to maximize the errors), how does the proposed method compare to other variance estimation methods, say what are the advantages and disadvantages?

**Limitations:**

NA.

---

> ### Author Rebuttal · Authors · 2024-08-03
>
> Thank you for your comments! Below, we address some of the points raised in your review.
>
> [Weakness 1] We have conducted an experiment in the attached pdf, and the details are in the global response to all reviewers.
>
> [Question 1] Most other variance estimation procedures in the literature (beyond the simple maximum-based estimator and the naive second moment estimator discussed in the experiment) require assumptions about the structure of the means; for example, the means are distributed according to a $k$-atomic distribution, spike-and-slab, etc. However, these structural assumptions can be unrealistic in some applications. The advantage of our method is that the means need only be assumed bounded.
>
> The disadvantage of our cumulant-based method is that it cannot, in its current form, adapt to any underlying structure of the $\mu$'s. If the means are truly structured, previous estimators can achieve faster convergence rates than the proposed cumulant-based estimator. As you point out, the choice $r \asymp \frac{\log n}{\log \log n}$ in our estimator is really made to protect against the worst case choice of $\mu$'s. It is an interesting question whether $r$ could be chosen in a data-driven way to adapt to the underlying structure of the $\mu$'s; this way, nothing but boundedness is assumed, yet existing structure could be exploited. At the current moment, it is not clear to us how to construct an adaptive version of our estimator; it is a nice direction for future work.

---

### Official Review · Reviewer_uxch · 2024-07-30

**Soundness:** 3
**Presentation:** 3
**Contribution:** 2
**Rating:** 6
**Confidence:** 3

**Summary:**

The paper studies the variance estimation of the normal means model and establishes the minimax squared error in terms of $n$, the number of observations. The results assume a bounded parameter space, where the absolute means and variance are at most 1 and $L^2$ (a large hidden constant), respectively. The estimator achieves the optimal rate through cumulant estimation.

**Strengths:**

- The major strength is determining the exact min-max rate regarding sample size, which requires both deriving a concrete estimator and establishing a lower bound (through moment matching).
- The results hold under relatively weak conditions, i.e., the boundedness of the parameter space. In comparison, prior research makes assumptions like smoothness, which might be less practical.
- The use of cumulants on top of the Gaussian character of the noise is innovative and might be of independent interest.

**Weaknesses:**

- The results show only the dependence on $ n$ but not $ L$, the upper bound on $ sigma$. Given that the optimal rate in $n$ is only inverse logarithmic, it seems important to understand if there are any gaps in $L$ between the lower and upper bounds (though it's treated as a universal constant).
- The boundedness assumptions are weaker, but Gaussianity is critical for establishing the paper's results. From this perspective, the methods may not be as widely applicable as some prior ones.
- The paper lacks numerical experiments. It might be helpful to run the estimator on synthetic data to understand the hidden constants and the estimator's actual performance.

**Questions:**

- Please feel free to clarify/respond regarding the above comments.
- Would it be possible to go slightly beyond Gaussianity and make the results more broadly applicable?

**Limitations:**

The comparisons are detailed, and the authors explain the limitations of their results. The paper is mostly theoretical, so potential negative societal impact may not apply.

---

> ### Author Rebuttal · Authors · 2024-08-03
>
> Thank you for your helpful feedback! Our responses below address the points raised in your review.
>
> [Weakness 1] Please see the global response to all reviewers regarding the dependence of the rate on $L$.
>
> [Weakness 2 and Question 2] We completely agree with your comment that Gaussianity is critical to the results. Indeed, the applicability is consequently limited. It is not at all clear to us how to generalize the cumulant methodology, and it would be an interesting, challenging direction for future work. The key property we use is that the variance of Gaussian noise only shows up in the second cumulant; this enables us to state a result like Proposition 1 to identify the noise variance from the marginal cumulants. For a different noise distribution, a different/analogous version of Proposition 1 needs to be established as the noise variance can now appear in other cumulants. The difficulty is reminiscent of the challenges of moment-based approaches discussed in Remark 2.
>
> [Weakness 3] We have conducted an experiment in the attached pdf, and the details are in the global response to all reviewers.

---

### Author Rebuttal · Authors · 2024-08-03

We thank all of the reviewers for their helpful comments. Below, we address those points which were raised by multiple reviewers.

Two reviewers asked about the dependence of $L$ in the minimax rate. Thank you both for the comment; it is well-received. We would like to make the case that $L \asymp 1$ addressed in the article is essentially the only interesting setting with a gap in the literature. In the small $L$ case where $L \lesssim \frac{1}{\sqrt{\log n}}$, it can be shown quite easily that the minimax rate is $L^4$ and is achieved by the trivial estimator $\hat{\sigma}^2 = 0$. Of course, one can then ask to capture the rate dependence on $L$ in the intermediate setting $\frac{1}{\sqrt{\log n}} \lesssim L \lesssim 1$, but we feel this case is of very limited interest and relevance. On the other side, the large $L \gtrsim 1$ case can be essentially addressed by existing literature. To elaborate, note one can transform the data by $Y_i = X_i/L$ which then is distributed as $Y_i \sim N(\theta_i, \tau^2)$ where $\theta_i = \mu_i/L$ and $\tau^2/L^2$. It is clear then that $\tau^2/L^2 \leq 1$ and $||\theta||\_\infty \leq L^{-1}$. To estimate $\sigma^2$, we can equivalently estimate $\tau^2$ and then rescale. Since $L$ is large, intuitively the nuisance $\theta_i$ should not have much effect, and so we can directly use $\hat{\tau}^2 = \frac{1}{n} \sum_{i=1}^{n} Y_i^2$. It is straightforward to show $E(|\hat{\tau}^2 - \tau^2|^2) \lesssim n^{-1} + L^{-4}$. It turns out this simple estimator can actually be optimal for estimating $\tau^2$. For example, in the scaling $L = n^\alpha$ where $\alpha > 0$ is fixed, $\hat{\tau}^2$ achieves the rate $n^{-1} + n^{-4\alpha}$, which is well known to be the minimax rate for estimating $\tau^2$, e.g. it can be seen as a consequence of reference [61]. Consequently, the rescaled estimator $\hat{\sigma}^2 = L^2\hat{\tau}^2$ must also be minimax rate optimal for $\sigma^2$. Therefore, in our view, essentially the only interesting case where new methodology is needed is $L \asymp 1$. We had thus focused our article on this setting.


Three reviewers pointed out that an experiment would be helpful. We have conducted an experiment in the attached pdf. The implementation details are as follows. We considered $20$ different various sample sizes $n$ spread evenly on the log-scale between $e^{7}$ and $e^{10}$. For each sample size $n$, we ran $100$ simulations each for four different data-generating processes. In particular, we chose four different priors $\mu \sim G$ being the Rademacher distribution ($G = \frac{1}{2}\delta_{-1} + \frac{1}{2}\delta_1$), a rescaled Beta distribution ($\mu = 2q - 1$ where $q \sim \text{Beta}(5, 1)$), the uniform distribution ($G = \text{Uniform}[-1, 1]$), and a Gaussian prior ($G = N\left(0, \frac{\log\log n}{\log n}\right)$). Note though the Gaussian prior is not supported on $[-1, 1]$, it is informative to consider since it is quite close to (more precisely, matches many moments with) the prior constructed in the lower bound argument of Section 3. For each choice of prior $G$, we sampled $\mu_1,...,\mu_n \overset{iid}{\sim} G$, and generated data $X_i \,|\, \mu_i \sim N(\mu_i, \sigma^2)$ with $\sigma = 2.25$. We computed the maximum-based estimator $\hat{\sigma}^2\_{\max} = \left(\max\_{1 \leq i \leq n} \frac{X_i}{\sqrt{2\log n}}\right)^2$, the naive second-moment estimator $\hat{\sigma}\_{\text{mom}}^2 = \frac{1}{n} \sum\_{i=1}^{n} X\_i^2$, and our cumulant-based estimator $\hat{\sigma}^2$ with the choice of tuning parameters $r = \left\lfloor \frac{\log n}{\log\log n}\right\rfloor$ and $\varepsilon = n^{-0.45}$. We computed the square error for each estimator, then computed the average square error across the $100$ simulations; these are the dots plotted in the figure. The error bars in the plot are standard errors.

From the experimental results, we can see that our cumulant-based estimator performs consistently well no matter the prior specification. In contrast, the maximum-based estimator (which achieves the upper bound $\frac{1}{\log n}$ as pointed out by reference [63]) performs consistently worse than the cumulant-based estimator across all priors. The behavior of the mean square error of the naive second-moment estimator is unsurprising, namely it related to $(E(\mu^2))^2$. Consequently, it does well in the Gaussian and Uniform prior settings where this quantity is low, but worse in the other settings where it is higher.

---

### Decision · Program_Chairs · 2024-09-25

**Decision:**

Accept (poster)

**Comment:**

The paper provides a sharp minimax rate for the variance estimation for the normal means model.

All reviewers were supportive of accepting the paper.

For a camera-ready version, please take into the account the comments from all reviewers regarding clarifications, for instance: inclusion of experiments, more discussion about the Gaussianity assumption and bounded means as compared to literature, and consider changing the title.